# Last Glacial ice sheet dynamics offshore NE Greenland – a case study from Store Koldewey Trough

Ingrid L. Olsen[1], Tom Arne Rydningen[1], Matthias Forwick[1], Jan Sverre Laberg[1], Katrine Husum[2]

[1]Department of Geosciences, UiT The Arctic University of Norway, Box 6050 Langnes, NO-9037 Tromsø, Norway
[2]Norwegian Polar Institute, Box 6606 Langnes, NO-9296 Tromsø, Norway

*Correspondence to:* Ingrid L. Olsen (ingrid.l.olsen@uit.no)

**Abstract**

The presence of a grounded Greenland Ice Sheet on the northeastern part of the Greenland continental shelf during the Last Glacial Maximum is supported by new swath bathymetry and high-resolution seismic data, supplemented with multi-proxy analyses of sediment gravity cores from Store Koldewey Trough. Subglacial till fills the trough, with an overlying drape of maximum 2.5 m thick glacier proximal and glacier distal sediment. The presence of mega-scale glacial lineations and a grounding zone wedge in the outer part of the trough, comprising subglacial till, provides evidence of the expansion of fast-flowing, grounded ice, probably originating from the area presently covered with the Storstrømmen ice stream and thereby previously flowing across Store Koldewey Island and Germania Land. Grounding zone wedges and recessional moraines provide evidence that multiple halts and/or readvances interrupted the deglaciation. The formation of the grounding zone wedges is estimated to at least 130 years, whilst distances between the recessional moraines indicate that the grounding line locally retreated between 80 to 400 meters/year during the deglaciation, assuming that the moraines formed annually. The complex geomorphology in Store Koldewey Trough is attributed to the trough shallowing and narrowing towards the coast. At a late stage of the deglaciation, the ice stream flowed around the topography on Store Koldewey Island and Germania Land, terminating the sediment input from this sector of the Greenland Ice Sheet to Store Koldewey Trough.

## 1    Introduction

The Greenland Ice Sheet (GIS) is the second largest ice sheet on Earth storing 2.9 million $km^3$ of ice (Dahl-Jensen et al., 2009). The GIS has experienced increasing ice loss during the last decades, contributing $0.6 \pm 0.1$ mm/yr to global sea-level rise between 2000-2010 (Fürst et al., 2015). About 16% of the GIS is presently drained via marine terminating outlet glaciers in NE Greenland, mostly through the North East Greenland Ice Stream (Joughin et al. 2000) consisting of three main outlets: 79°-Glacier, Zachariae Isstrøm and Storstrømmen (e.g. Rignot and Kanagaratnam 2006) (Fig. 1). A future warming global climate, which will be particularly strong in the Arctic (Serreze and Francis, 2006), will possibly lead to a reduced sea-ice cover adjacent to the glacier termini and subsequent accelerated melting of the ice sheet in NE Greenland (Bendtsen et al., 2017). This could cause an instability and possibly irreversible loss of the GIS, which has - together with the West-Antarctic Ice Sheet (WAIS) - been identified as a tipping element in the Earth's climate system (Lenton et al., 2008). A complete meltdown of these ice sheets can potentially lead to a global sea-level rise of 7.3 (GIS) m and 3.2 m (WAIS) (Bamber et al., 2001, 2009), causing severe consequences for coastal societies (IPCC, 2018). However, precise predictions of future evolvement of the GIS remains difficult (Nick et al., 2013). A better understanding of the development of glaciers in response to past climate changes, e.g. from the Last Glacial Maximum (LGM; c. 26.5-19 ka BP (Clark et al., 2009)) towards the present, is needed to validate and improve numerical models focusing on present processes, as well as the future development of glaciers and ice sheets.

The reconstruction of the GIS configuration and dynamics from marine-geoscientific data, including maximum extent during the LGM, as well as the timing and dynamics of the deglaciation, have been addressed in multiple studies (e.g. Andrews et al., 1998; Bennike et al., 2002; Dowdeswell et al., 1994, 2014; Funder et al., 2011a; Hogan et al., 2011, 2016, 2020; Hubberten et al., 1995; Ó Cofaigh et al., 2013). However, these reconstructions focus primarily on the southern and western sectors offshore Greenland, and reconstructions from offshore NE Greenland remain sparse (Arndt, 2018; Arndt et al., 2017; Evans et al., 2002; Laberg et al., 2017; Stein et al., 1996; Winkelmann et al., 2010). Marine geoscientific studies suggest that the northeastern sector of the GIS extended all the way to the shelf edge during the last glacial based on observations of subglacial and ice-marginal depositional landforms, including mega-scale glacial lineations, grounding zone wedges and recessional terminal moraines (Arndt, 2018; Arndt et al., 2015, 2017; Evans et al., 2009; Laberg et al., 2017; Ó Cofaigh et al., 2004).

Laberg et al. (2017) presented glacial landforms interpreted as retreat moraines in the outermost part of Store Koldewey Trough (Fig. 1), suggesting a stepwise early deglaciation likely triggered by an increase in ocean temperature. However, in order to precisely link retreat events with external forcing (e.g. climate or oceanic changes), an absolute chronology for the deglaciation is required. According to Evans et al. (2002), breakup and retreat of the GIS further to the south, outside Kejser Franz Joseph Fjord (for location, see Fig. 1), commenced after 18.4 cal. ka BP (for age calibration see Material and methods chapter), with the ice abandoning the mid-shelf before 15.5 cal. ka BP and the inner shelf being ice free before 10.3 cal. ka BP. Cosmogenic nuclide dating on Store Koldewey Ø, located west of Store Koldewey Trough (Fig. 1), reveals that the ice front retreated from the area c. 12.7 ka BP (Skov et al., 2020), whereas the ice front rested east of the present coastline of Germania Land until c. 10 cal. ka BP (Landvik, 1994). By 8.3 cal. ka BP the ice front had retreated close to its present position, and after further recession Germania Land became an island about 5.5 cal. ka BP. Storstrømmen readvanced again c. 1 cal. ka BP, reaching its present position during the Little Ice Age (Weidick et al., 1996).

The overall objective of this paper is to provide new knowledge about the evolution of the northeastern part of the GIS based on new acoustic data (multibeam bathymetry and Chirp seismic) and sediment cores. These new data sets expand and complement existing data in one of the largest glacial troughs offshore NE Greenland, i.e. the Store Koldewey Trough. More specifically, the aims are to 1) reconstruct the ice drainage pathways, ice sheet extent and ice stream dynamics of this sector of the GIS overlying Store Koldewey Trough during the LGM and the deglaciation, and 2) discuss the post-glacial marine environmental conditions of Store Koldewey Trough.

## 2    Regional setting

The large-scale morphology of the NE Greenland continental shelf is characterized by several large cross-shelf troughs separated by shallower banks and shoals (Fig. 1). The troughs are characteristic features of formerly glaciated continental shelves, interpreted as glacially over-deepened landforms acting as conduits for fast-flowing ice streams eroding into the sub-glacial bed (e.g. Vorren et al. 1988; Canals et al. 2000; Batchelor and Dowdeswell 2014), whilst inter-trough banks are interpreted to have been covered by slower flowing ice, consequently experiencing less erosion (Klages et al., 2013; Ottesen and Dowdeswell, 2009).

The east coast of Greenland is presently largely influenced by the southward flowing East Greenland Current carrying cold, fresh surface Polar Water and sea-ice from the Arctic Ocean together with warmer modified Atlantic Intermediate Water (Aagaard and Coachman, 1968; Hopkins, 1991). An increased inflow of warm Atlantic Intermediate Water into East Greenland troughs and fjords is proposed to influence the submarine melt rates, causing an instability at the grounding line of marine terminating outlet glaciers (Khan et al., 2014; Mayer et al., 2018), e.g. the 79°-Glacier (Straneo and Heimbach, 2013; Wilson and Straneo, 2015).

Store Koldewey Trough is a ~210 km long, 30-40 km wide and up to 400 m deep, NW-SE oriented cross-shelf trough located at ~76° N offshore NE Greenland (Fig. 1). It is divided into an inner-, middle- and outer trough (Fig. 1C), with a sinuous centerline terminating at the shelf edge. A bathymetric profile along the axis of Store Koldewey Trough reveals that it differs from most troughs offshore NE Greenland, because it overall deepens

towards the shelf break (Fig. 1B). In addition, Store Koldewey Trough is the only trough on the NE Greenland continental shelf without a fjord continuation; it terminates near Germania Land and the island Store Koldewey to the west (Fig. 1A).

5    Laberg et al. (2017) identified an assemblage of glacigenic landforms in outer Store Koldewey Trough, including mega-scale glacial lineations and rhombohedral- and transverse ridges with variable dimensions. From the landform assemblage, it was inferred that grounded ice expanded to the shelf edge during the last glacial. Four prominent transverse ridges located in the outer, middle and inner trough were interpreted as grounding-zone wedges deposited in front of the GIS at temporary at stillstand and/or readvances during the last deglaciation (see 10    labels *A-D* in Fig. 1B, C). Similar landform assemblages are also identified in other troughs along the northeastern continental shelf of Greenland (Arndt, 2018; Arndt et al., 2015) as well as on other formerly glaciated continental shelves (e.g. Ottesen et al. 2005; Winsborrow et al. 2010; Jakobsson et al. 2012a; Bjarnadóttir et al. 2013; Andreassen et al. 2014; Batchelor and Dowdeswell 2014). However, details about how these shelf-break terminating parts of the North East Greenland Ice Sheet retreated landwards remain unknown, as only very limited 15    areas are mapped by high-resolution swath bathymetry (Arndt, 2018; Arndt et al., 2017; Evans et al., 2009; Laberg et al., 2017; Winkelmann et al., 2010).

### 3    Material and methods

Acoustic data, including swath bathymetry and high-resolution seismic data, as well as four sediment gravity cores 20    were collected during cruises arranged by the TUNU-program (Christiansen, 2012) using *R/V Helmer Hanssen* of UiT The Arctic University of Norway in 2013, 2015 and 2017.

The swath bathymetry data were acquired using hull-mounted Kongsberg Maritime Simrad EM 300 and 302 multibeam echo sounders in 2013/2015 and 2017, respectively. Sound velocity profiles for the water column were 25    derived from CTD (conductivity-temperature-depth) casts prior to and during the bathymetric surveys. High-resolution seismic profiles were acquired with a hull-mounted EdgeTech 3300-HM (Chirp) sub-bottom profiler simultaneously with the swath bathymetry data, using a pulse frequency of 2-8 kHz. Visualization and interpretation of the Chirp and swath bathymetry data were performed using Petrel 2018 and Global Mapper 19.

30    The acoustic data sets consists of previously unpublished data from the middle trough acquired in 2017, and data from the outer trough previously presented by Laberg et al. (2017). Systematic mapping and outlining of landform elements in Store Koldewey Trough were conducted on the entire swath bathymetry data set, including some re-interpretations of the outer trough (see subchapter *4.2.4. Curvilinear ridges – Saw-tooth recessional moraines*). The landforms were mapped and classified based on shape, size, arrangement and orientation. The sediment 35    volumes of the grounding zone wedges were calculated as trapezoid prisms, using a mean thickness and length obtained from the acoustic data. Due to the limited data coverage, we use the similar approach as Jakobsson et al. (2012a) and calculate the volume per 1-m grounding line width ($m^3$/m). Since the Chirp sub-bottom signal was unable to penetrate to the bases of the grounding zone wedges, a flat base beneath the proximal sides of the wedges was assumed.
40
The sediment gravity cores (HH17-1326, HH17-1328, HH17-1331 and HH17-1333) were retrieved from 294 m to 345 m water depth along a transect extending from inner to middle Store Koldewey Trough using a 6 m long steel barrel (Fig. 1; Table 1). Coring sites were chosen with the purpose of penetrating a stratigraphic sequence including subglacial and glacimarine deposits.
45
Prior to opening, the physical properties of the sediments were measured using the GEOTEK Multi Sensor Core Logger, with a 10 mm step size and 10 s measuring time. The cores were stored for one day in the laboratory prior to the measurements to allow the sediments to adjust to room temperature as temperature changes can affect the physical properties (Weber et al., 1997). After splitting, color images were acquired with a Jai L-107CC 3 CCD 50    RGB Line Scan Camera installed on an Avaatech XRF core scanner. Furthermore, X-radiographs were taken with

a GEOTEK MSCL-XCT X-ray core imaging system. A systematic description of the sediment surface was carried out and colors were determined visually using the Munsell Soil Color Chart (Munsell, 2000). The cores contained insufficient amounts of dateable material for a chronology to be established, a common problem from areas where cold polar waters lead to both relatively little calcareous organisms and dissolution of the carbonate material (e.g. Andrews et al., 1999; Zamelczyk et al., 2012).

Grain-size analyses were performed using a Beckman Coulter LS 13 320 laser particle size analyzer, measuring the range from 0.04 µm to 2000 µm. Particles larger than 2000 µm were removed by a sieve and are presented as clasts in the lithological logs. Prior to the analyses, chemical treatment of the samples using HCl and $H_2O_2$ was conducted to remove carbonates and organic content, respectively. Distilled water was added to the samples before being shaken for 24 hours. Furthermore, two drops of Calgon solution were added to the samples before being placed in an ultrasound bath for five minutes in order to disintegrate aggregates of particles. Each sample was analyzed three times and the particle size statistics were calculated using GRADISTAT v. 8.0 (Blott and Pye, 2001).

Age calibration for the radiocarbon ages cited in this study was performed using the CALIB 8.20 software (Stuiver et al., 2020), applying the Marine20 calibration curve (Heaton et al., 2020) with $\Delta R = 0 \pm 50$, as recommended by Andrews et al. (2016). All dates are presented in calibrated years before present (cal. ka BP), where zero year BP is AD 1950.

## 4    Results

### 4.1    Lithostratigraphy of the uppermost trough strata

Five lithofacies are defined based on the lithological composition, sedimentary structures and physical properties (Fig. 2 and 3). The properties of the different facies are summarized in Table 2.

#### 4.1.1  Facies 5 - Diamicton (Dmm)

The lowermost facies in all four cores comprises a very dark gray, massive, matrix-supported diamicton with a sandy mud matrix and high amounts of randomly oriented clasts (Fig. 2 and 3; Table 2). The upper boundary is sharp, and bioturbation is absent. The magnetic susceptibility varies between each core, with the highest in HH17-1326 and lowest in HH17-1328. The wet bulk density is generally higher than in the overlying facies, suggesting over-consolidation of the sediments.

Based on the diamictic composition of these deposits, the high amounts of clasts, absence of bioturbation and a considerable consolidation of the sediments we suggest that the facies represents diamictic subglacial debris/basal till deposited at the base of an ice stream from the GIS (compare with Evans et al. 2006).

#### 4.1.2  Facies 4 – Interlaminated glacimarine sediments (Fl)

Facies 4 is present in the two cores in the inner trough; HH17-1326 and HH17-1328, with thicknesses of 12 cm and 35 cm, respectively (Fig. 2 and 3, Table 2). The facies consists of dark gray laminated mud with fine sandy layers. Bioturbation and clasts are absent. The upper unit boundary is gradational and defined by the appearance of clasts. The wet bulk density is medium, whilst the magnetic susceptibility varies with a decrease in HH17-1326 and an increase in HH17-1328 relative to the underlying unit.

Facies 4 is interpreted to contain glacier-proximal glacimarine sediments deposited from suspension settling, where the stratification of the sediments reflects variations in the current strength of the water masses emanating from the ice margin. Such laminated to massive sandy muds are typically overlying basal diamictons (i.e. facies

5) in glaciated continental margin and fjord settings (e.g. Domack et al., 1999; Prothro et al., 2018; Smith et al., 2019). The lack of clasts interpreted as ice-rafted debris (IRD) in ice-proximal settings may have several explanations, which will be discussed further below (see chapter 5.2).

### 4.1.3 Facies 3 – Interlaminated glacimarine sediments with occasional IRD (Fl (d))

Facies 3 occurs in all cores. It is 9-23 cm thick and consists of dark gray laminated mud with fine-sand layers and clasts (Fig. 2 and 3; Table 2). The clasts, interpreted as IRD, appear in diamict layers. The unit has a sharp or gradational lower boundary overlying either facies 5 or 4, respectively, whilst the upper boundary is gradational. The properties are similar to facies 4, with medium-high wet bulk density. However, the magnetic susceptibility varies between the cores.

Facies 3 is interpreted to represent a glacier-proximal setting with glacimarine sediments containing IRD deposited in episodic calving events. The sediments are thought to have been deposited in glacimarine conditions, reflecting a calving zone in a more distal setting where IRD becomes a more dominant component at the expense of suspension settling (cf. Smith et al., 2019), compared to facies 4. Another possible explanation is that the facies reflects deposition from melt out of englacial debris at the grounding line, similar to the facies ascribed as ´stratified diamicton` by Smith et al. (2019).

### 4.1.4 Facies 2 – Massive glacimarine sediments (Fm)

Massive olive gray to dark gray mud with little to moderate bioturbation and rare clasts composes facies 2. The facies is 10-80 cm thick and occurs in all four sediment cores (Fig. 2 and 3; Table 2); facies 2 overlies facies 3 in all cores, except for core HH17-1331, where it overlies facies 1. In core HH17-1333, facies 2 alternates with facies 1. The lower and upper unit boundaries are gradational in all cores. The physical properties, including wet bulk density and magnetic susceptibility vary slightly within the facies and between the cores.

The facies is interpreted to reflect suspension settling in an ice-distal glacimarine environment with limited iceberg or sea-ice rafting. With increased distance from the grounding line, deposition from turbid meltwater plumes typically grade into more massive, bioturbated mud (Ó Cofaigh and Dowdeswell, 2001; Prothro et al., 2018; Smith et al., 2019). The rare amount of IRD within the massive mud may be a consequence of the appearance of warm surface water during the Early Holocene causing prolonged open water conditions and limited iceberg rafting on the shelf (Müller et al., 2012; Syring et al., 2020).

### 4.1.5 Facies 1 – Massive glacimarine sediments with IRD (Fm (d))

The uppermost facies in all of the studied cores consists of massive mud with clast-containing intervals, the latter interpreted to be IRD (Fig. 2 and 3; Table 2). Facies 1 occurs twice in core HH17-1331 and HH17-1333, lying directly above facies 3 and 2. Sediment color alternates between brown to dark grayish brown, as well as olive gray to dark olive gray. Facies 1 is generally coarser than facies 2. Peaks in the magnetic susceptibility correspond to the depth with highest abundance of coarser material. The wet bulk density is similar to the underlying facies 2.

Facies 1 is interpreted to have been deposited in a similar environment as facies 2. Deposition of IRD can occur from dropping and dumping (see Vorren et al., 1983), i.e. dumping from a single iceberg or ice flow may be misinterpreted as enhanced ice rafting. However, since we identify increased amounts of IRD in all four cores we are confident that facies 1 reflects increased ice rafting at a regional scale, most probably related to the Neoglacial cooling trend (cf. Syring et al., 2020).

### 4.2 Submarine landforms: glacial – deglacial ice sheet dynamics

The swath bathymetry data from the middle (Fig. 4) and outer (Fig. 5) Store Koldewey Trough reveal glacigenic landforms interpreted to reflect various stages of ice sheet extent, flow dynamics and retreat patterns.

### 4.2.1 Streamlined landforms – Glacial lineations

Streamlined, trough-parallel grooves and ridges occur in the middle and outer trough (Fig. 4, 5 and 6A), terminating close to the shelf edge. Individual ridges have widths of 150-500 m and reliefs between 4-8 m (Table 3). They occur in clusters with spacing from 200 to 700 m. The grooves and ridges are partly eroded and/or overprinted by other landforms, making the determination of their maximum lengths challenging. Their minimum lengths range from 1.5 to >9 km long and their length/width ratios generally exceed 10:1, with the highest ratios occurring in the outer trough. High-resolution seismic data show that the ridges are acoustically transparent, a property that is characteristic for basal till (e.g. Ó Cofaigh et al., 2007).

Based on the spatial distribution, dimensions and orientations, we interpret the grooves and ridges as glacial lineations formed subglacially at the base of a fast-flowing, grounded ice stream (Clark, 1993; King et al., 2009; Spagnolo et al., 2014) draining the GIS towards the shelf break. The lineations in the outer trough have longer length/width-ratios and are more densely spaced, and are, thus, termed mega-scale glacial lineations. Similar streamlined landforms have been described on the seafloor of other formerly glaciated margins where they have been interpreted to indicate the presence of grounded ice streams (e.g. Canals et al. 2000; Ottesen et al. 2005; Evans et al. 2009; Rydningen et al. 2013; Andreassen et al. 2014; Hogan et al. 2016; Arndt 2018).

### 4.2.2 Large transverse ridges - Grounding zone wedges

Four prominent bathymetric sills, interpreted as grounding zone wedges *A-D* from multibeam data and IBCAO 3.0 (Jakobsson et al., 2012b) by Laberg et al. (2017), are present within the trough (Fig. 1B, C and 7). These authors presented acoustic data from wedges *A* and *B*, while the data from our study provides new information about wedge *C*. The wedges are 35-100 m high, 3.5-10 km wide and are spaced 45-60 km apart from each other (Table 3). Sediment volumes per meter grounding line width are approximately 130 000 $m^3$, 738 000 $m^3$ and 150 000 $m^3$ for wedges *A*, *B* and *C*, respectively. The cross-trough extents of the grounding zone wedges exceed the multibeam data coverage. Smaller ridges overprint the grounding zone wedges (Fig. 4, 5 and 6). The base of the wedges is impossible to identify on our Chirp profiles, but 2-D seismic profiles described by Petersen et al. (2015) reveal a thick Neogene sedimentary succession offshore NE Greenland, thus ruling out that these features are bedrock sills. As such, these large landforms are interpreted to be accumulations of sediments deposited at the grounding line of the ice stream, recording the position of the grounding line during temporary stillstand, either reflecting a halt during the general retreat or the termination of a more extensive readvance during a late phase of the last glacial.

### 4.2.3 Small transverse ridges – Transverse recessional moraines

Multiple straight to slightly curvilinear transverse/semi-transverse ridges are visible on the seafloor of Store Koldewey Trough (Fig. 4, 5, 6 and 7). The ridges are up to 2200 m wide, have reliefs <50 m and have a spacing of 50-500 m (Table 3). Some of the ridges superimpose others, implying several generations of ridge formation (Fig. 5). There are two generations of ridges in the outer trough (between grounding zone wedges *A* and *B*), where the first generation is spaced ~80 m apart, whilst the superimposing ridges are larger and mostly spaced 200-400 m apart. The spacing of ridges in the middle trough is commonly 100-200 m.

We interpret the curvilinear to straight ridges as recessional moraines formed at the grounding line during overall retreat with repeated short-term stillstand and/or small readvances (cf. Dowdeswell et al., 2008; Ó Cofaigh et al., 2008). The ridge-like features identified on the sub-bottom profiles are acoustically transparent suggesting a diamictic composition (Stewart and Stoker, 1990) (Fig. 7).

### 4.2.4 Curvilinear ridges – Saw-tooth recessional moraines

Clusters of curvilinear ridges occur both landward and seaward of grounding zone wedge *B,* and seaward of grounding zone wedge *C* (Fig. 4, 5, 6B and 7D, E). These ridges occur often closely spaced, exhibiting a saw-tooth pattern in plan view. Many of the features continue as long moraine ridges oriented sub-parallel to the ice-flow direction. Bifurcations and cross-cutting patterns occur. Individual ridges are up to 1.3 km long, 170-1100 m wide and have a relief of 5-30 m (Table 3). They are typically asymmetrical with a steeper ice distal slope and a more gentle ice proximal slope (Fig. 6E). The saw-tooth ridges partly superpose and modify the underlying transverse ridges creating locally chaotic seafloor patterns. Furthermore, grounding zone wedge *B* partly covers some saw-tooth ridges (Fig. 6B).

We interpret the saw-tooth ridges as recessional moraines that formed by a combination of push- and squeeze processes, recording an active ice retreat punctuated by periodic advances. The formation of these distinctive landforms is inferred to be dependent on the topography, where down-ice widening, in this case of the trough, causes increased transverse stress leading to longitudinal crevasses initiating an irregular ice front. Similar saw-tooth like moraines have been observed in e.g. Norway (Burki et al., 2009; Matthews et al., 1979), Barents Sea (Hogan et al., 2010; Kurjanski et al., 2019), Iceland (Chandler et al., 2016; Evans et al., 2016) and Arctic Canada (Andrews and Smithson, 1966). The landforms were previously interpreted as rhombohedral ridges by Laberg et al. (2017) in the western part of the data set from the outer trough. However, we find the saw-tooth-like morphology incompatible with the geometric ridge networks of rhombohedral ridges based on the ridges zig-zag pattern in plan view (cf. Bennett et al., 1996).

### 4.2.5 Straight incisions - Channels

Two straight incisions that are U-shaped in cross section, 150-300 m wide and with incision depths of 3-10 m are identified on the northern and southern trough sidewalls (Fig. 4; Table 3). The incisions are oriented parallel to the recessional moraines and continue beyond the extent of the swath bathymetry data set. They cut into the glacial lineations and the acoustically transparent sediments interpreted as basal till. The landforms are interpreted as channels formed during deglaciation and are probably related to erosion by meltwater.

### 4.3 Seismostratigraphy

Two seismostratigraphic units (S1 and S2) are distinguished in the Chirp sub-bottom profiles in Store Koldewey Trough (Fig. 7).

### 4.3.1 Unit S1 – Glacigenic deposits and/or sedimentary bedrock

Unit S1 is the lower seismostratigraphic unit and the base of this unit represents the acoustic basement. The unit has an acoustically transparent to semi-transparent signature and an irregular top reflection with medium to high amplitude and continuity (Fig. 7B).

The unit correlates with lithological unit 5 (*Dmm*) in the sediment cores, interpreted as subglacial till, i.e. it includes subglacial deposits. However, the internal reflection shown in Fig. 7D can either be interpreted as a bedrock surface visible on the Chirp profile due to a thin layer of till, or as an internal reflection within the till. Furthermore, the Chirp profiles (Fig. 7) and sediment cores confirm that the grounding zone wedges and recessional moraines consists of subglacial till.

### 4.3.2 Unit S2 – Glacimarine sediments

Unit S2 is acoustically transparent (Fig. 7). The unit, maximum 2.5 m thick, occurs only locally either as an infill between the topographic highs or draping the underlying unit S1. It is absent in most of Store Koldewey Trough (e.g. in the outer trough).

Unit S2 is correlated with the lithological units 4 (*Fl*), 3 (*Fl (d)*), 2 (*Fm*) and 1 (*Fm (d)*), and it occurs at all four core sites. Thus, the unit contains glacimarine deposits reflecting a gradual transition from glacier proximal to distal environments.

## 5    Discussion

### 5.1    Maximum ice sheet extent and influence of subglacial topography

Mega-scale glacial lineations extending almost to grounding zone wedge *A* in the outer trough (Fig. 5 and 9: Stage 1), together with subglacial debris/basal till in sediment cores from the middle trough, suggest that a grounded, fast-flowing ice stream draining the northeastern sector of the GIS extended to the shelf break in Store Koldewey
Trough at maximum ice extent during the last glacial (Fig.8). Furthermore, acoustic profiles reveal an up to 2.5 m thick drape of glaciomarine sediment overlying the subglacial till in certain parts of the inner and middle trough, whereas a detectable sediment drape in the outer trough is absent (Fig. 5). This lack of glacimarine sediments and good preservation of glacial landforms in the outer trough indicate that the identified landforms formed during the LGM and the subsequent deglaciation, as proposed by Laberg et al. (2017).

The shelf break-terminating ice stream in Store Koldewey Trough is consistent with reconstructions of shelf-break terminating glaciers during the LGM elsewhere on the NE Greenland Margin, i.e. ranging from our study area in the south to the Westwind Trough at 80.5° N in the north (Arndt et al., 2015, 2017; Laberg et al., 2017; Winkelmann et al., 2010) (Fig. 1A). If the maximum glacier extent on the margin occurred synchronously, this
implies that an ice sheet front covered a minimum length of 680 km along the outer shelf.

Stein et al. (1996) presented a chronology of the deposition of terrigenous, coarse grained material along the continental slope off NE Greenland, suggesting that the maximum late Weichselian ice extent occurred at about 24-19 cal. ka BP. Radiocarbon dates from the Greenland Basin indicate that mass-wasting activity in a channel
system on the upper continental slope took place predominantly under full glacial and deglacial conditions, and that this had ceased after about 14.7 cal. ka BP, leaving the channels largely inactive (Ó Cofaigh et al., 2004). Thus, from the data available, the outer parts of Store Koldewey Trough may have been ice covered in the period from ~24-14.7 cal. ka BP.

We propose that the Store Koldewey Trough was filled by grounded ice masses deriving from the area presently covered by the Storstrømmen ice stream (Fig. 8A). Palaeo-ice sheet models have calculated that the ice covering Germania Land during LGM reached 1000-1500 m ice thickness (Fleming and Lambeck, 2004; Heinemann et al., 2014). This implies that the northeastern sector of the GIS likely reached a thickness allowing the ice stream to flow across the underlying topography, including the mountain range with 500-900 m high peaks between present
day Storstrømmen and Germania Land. Such "pure" ice streams (Bentley, 1987; Stokes and Clark, 1999), flowing unrelated to topography, are documented from the contemporary Siple Coast Ice Streams of West Antarctica. Moreover, the disregard for topography appears to be a characteristic of both the palaeo and contemporary North East Greenland Ice Stream (Fahnestock et al., 1993; Sachau et al., 2018) (Fig. 1A), as well as for the Maskwa paleo-ice stream within the Laurentide Ice Sheet (Ó Cofaigh et al., 2010; Ross et al., 2009).

An alternative interpretation is that Store Koldewey Trough had a much smaller drainage-basin, limited to Germania Land, as proposed by Arndt et al. (2015). However, based on our data, including the observations of mega-scale glacial lineations, recessional moraines and grounding zone wedges, as well as the estimates from Fleming and Lambeck (2004) and Heinemann et al. (2014), we favor the interpretation that the Storstrømmen ice
stream sourced Store Koldewey Trough during full glacial conditions, based purely on the volume of ice needed

to fill a trough of this dimension. Germania Land covers an area of ~2500 km$^2$, and if the ice thickness here reached 1000-1500 m during LGM (Fleming and Lambeck, 2004; Heinemann et al., 2014), the total ice volume must have been 2500-3750 km$^3$. Store Koldewey Trough covers an area of ~9000 km$^2$, and the present-day water depth at the shelf break is >400 m. Thus, the volume needed to fill the trough exceeds the ice volume estimated for Germania Land, and a local drainage basin from there is therefore unlikely.

## 5.2 Ice stream dynamics during deglaciation

The complex glacial landform assemblage in Store Koldewey Trough, comprising transverse wedge- and ridge systems, reflects to a large degree the dynamic retreat of the ice margin during the deglaciation. The types of landforms and their spatial distributions can be attributed to the overall seafloor topography of the trough, with a seaward dipping bed slope, supplemented by local pinning points related to trough shallowing and/or narrowing. The resulting deglacial dynamics was characterized by several periods of stabilization and readvances of the grounding line in Store Koldewey Trough during overall retreat. In contrast, many paleo-ice streams on other glaciated continental shelves with landward dipping beds have experienced a lift-off from the seafloor and an initial rapid retreat due to sea-level rise, e.g. Norske Trough (Arndt et al., 2017), Amundsen Sea in West Antarctica (Smith et al., 2011) and NW Fennoscandian Ice Sheet (Rydningen et al., 2013).

The retreat landforms of various morphologies mark different retreat styles and periods of grounding line stabilization during retreat (Fig. 9: Stage 3 and 4). Whilst the spatial distribution of the moraine ridges indicates a stepwise retreat with several episodes of relatively short grounding line stabilizations, the presence of four large grounding zone wedges indicate that the glacier also grounded for a longer time during the deglaciation, allowing for larger wedges to form (Dowdeswell et al., 2008; Ó Cofaigh et al., 2008). Because the formation of grounding zone wedges and recessional moraines require a grounded ice stream/glacier margin we exclude a rapid/continuous retreat by ice stream flotation.

The formation of grounding zone wedges typically requires a stabilization of the ice margin for decades to centuries (Dowdeswell and Fugelli, 2012) (Fig. 9: Stage 4). This period can be estimated when sediment flux across the grounding line and grounding zone wedge volume are known (Howat and Domack, 2003). Grounding zone wedges A to C in Store Koldewey Trough have volumes of approximately 130 000 m$^3$, 738 000 m$^3$ and 150 000 m$^3$ per meter grounding line width. In the absence of chronology we apply a sediment flux of $10^2$ to $10^3$ m$^3$ m$^{-1}$ yr$^{-1}$ to the grounding line, as calculated for other paleo ice streams in Greenland (Hogan et al., 2012, 2020), the Whillans Ice Stream, West Antarctica (Anandakrishnan et al., 2007), and on the southern Norwegian continental margin (Nygård et al., 2007). Applying these upper and lower sediment flux rates, the periods required for the formations of grounding zone wedges A, B and C are in the range of 130-1300, 740-7400 and 150-1500 years, respectively. The lower flux rate, resulting in an order of magnitude longer formation time, seems less likely given the tight time frame available for retreat across the shelf (~14.7-12.7 ka BP (Ó Cofaigh et al., 2004; Skov et al., 2020)).

The recessional moraines are generally one to three orders of magnitude smaller than the grounding zone wedges. Accumulations of retreat moraines have repeatedly been referred to as 'annual moraines' correlated with annual cycles including winter advances and summer retreats during the overall deglaciation (Baeten et al., 2010; Boulton, 1986; Kempf et al., 2013; Ottesen and Dowdeswell, 2006) (Fig. 9: Stage 3). Whilst annual formation of moraines can be studied in situ in e.g. Svalbard (Flink et al., 2015; Ottesen and Dowdeswell, 2006), the interpretation of a series of evenly spaced recessional moraines as annually in the palaeo-record is debated (e.g. Chandler et al., 2020). However, assuming that accumulations of retreat moraines reflect annual moraines, we propose the following deglaciation velocities in the study area: following the formation of grounding zone wedge A at the shelf edge, the grounding line 1) retreated with an average of 80 m yr$^{-1}$, 2) readvanced and 3) retreated again, accelerating to 200-400 m yr$^{-1}$ in the outer trough (Fig. 9: Stage 6). By the time the ice margin reached mid-trough, spacing of individual moraines indicate a reduced recession of 100-200 m yr$^{-1}$. Although estimates on sediment flux and deglacial rates are presented here, we recognize that there are uncertainties in our calculations. Therefore, more precise calculations remain to be defined from other data than ours.

The lithological sequence starting with a basal till overlain by glacimarine deposits suggests the transition from sub-glacial to an ice-proximal setting and, subsequently, to a more ice-distal environment dominated by suspension settling and various degrees of ice rafting. The deglacial lithofacies (3 and 4) reflect different depositional environments (Table 2): whereas the influence from meltwater was stronger during the deposition of facies 4, the supply of IRD was higher during the deposition of facies 3. The lack of IRD in an ice proximal setting may have several explanations; i) the time of deposition may represent a period with an extensive sea-ice cover preventing melt-out of debris in the area (Jennings and Weiner, 1996; Moon et al., 2015; Vorren and Plassen, 2002), ii) a high flux of sediment-laden glacial meltwater masks the amount of IRD (Boulton, 1990) or iii) the sediments may be deposited in a sub-shelf environment far enough from the grounding line to be unaffected by mass flows and rain-out of basal debris at the grounding line (Domack and Harris, 1998; Reilly et al., 2019; Smith et al., 2017).

We note that facies 4, characterized by lamination and the absence of clasts, occurs exclusively in the two cores in the inner trough. Given that the coring sites are located within depressions, it could be assumed that the ice detached from the ground leading to sub-ice shelf environments where deposition was dominated by suspension settling. We speculate that the trough narrowing towards the coast contributed to an increase in lateral drag and subsequent reduction in extensional stress as the ice front retreated to the inner trough, resulting in a more stabilized ice front and ice-shelf formation (Fig. 9: Stage 5). Trough narrowing towards the coast could possibly have contributed to an increase in lateral drag and subsequent reduction in extensional stress as the ice front retreated to the innermost trough area, resulting in a more stabilized ice front (grounded) and ice-shelf formation from here (Fig. 9: Stage 5). Thus, facies 4 deposited in the inner trough while the glacier was grounded further to the west, in a similar setting as described by Reilly et al. (2019) for the Petermann Glacier in NW Greenland, where an IRD free depositional environment beneath the floating ice tongue was followed by an increase in IRD concentrations as the ice tongue retreated from the site.

### 5.3 Postglacial development

During the late phase of the deglaciation, the flow path of the ice stream became controlled by the topography on Store Koldewey Island and Germania Land, i.e. it was directed into Jøkelbugten to the north and Dove Bugt in the south (Fig. 8B and 9: Stage 6). This terminated the supply of suspended sediments and icebergs from the Storstrømmen area to Store Koldewey Trough. The generally thin and patchy occurrence of seismostratigraphic unit S2 (correlating to facies 1-4; Fig. 7) could be due to a change of ice configuration in the west, with ice masses eventually being routed north- and southwards, resulting in a decrease of sediment supply to the trough. More specifically, the absence of a detectable post-glacial sediment cover in the outer trough may be a result of ocean current winnowing from the southward-flowing East Greenland Current, leading to non-preservation of fine-grained sediments here.

Postglacial sedimentary processes in the trough are interpreted to comprise hemipelagic deposition of terrigenous material from sea-ice transported across the Arctic Ocean within the Transpolar Drift, rafting by sea-ice formed along the NE Greenland coast, rainout from icebergs and meltwater plumes released from regional marine terminating outlet glaciers north of the study area (e.g. 79°-Glacier and Zachariae Isstrøm), as well as winnowing on the surrounding banks.

The low IRD content in facies 2 is probably due to multiple factors: i) ice fronts retreating on land, ii) material entrapped in icebergs calving off from marine-terminating glaciers probably melted out rapidly and icebergs only occasionally reached the continental shelf. This could correlate to the Holocene Thermal Maximum (c. 8-5 ka BP) when temperatures in NE Greenland were higher than at present (e.g. Dahl-Jensen et al., 1998; Klug et al., 2009), causing the Storstrømmen ice margin to retreat behind its present ice-extent (Bennike and Weidick, 2001; Weidick et al., 1996). Furthermore, sea-ice formation in the Arctic Ocean and on the NE Greenland shelf was reduced during that period (Koç et al. 1993; Funder et al. 2011b; Müller et al. 2012; Werner et al. 2016). The increasing input of IRD towards the top of the sediment cores (Fig. 2) is attributed to the subsequent regional climatic cooling.

This climate deterioration, referred to as the Neoglaciation (c. 5 ka BP – Little Ice Age), led to glacier expansion with enhanced iceberg-rafting and increased sea-ice extent on the East Greenland shelf (Klug et al., 2009; Müller et al., 2012).

**6 Conclusions**

- New and previously published swath bathymetry data (Laberg et al., 2017), integrated with high-resolution seismic data and sediment gravity cores, provide new information about the dynamics of the northeastern sector of the Greenland Ice Sheet, as well as about glacial and postglacial sedimentary environments.
- The lithostratigraphy in Store Koldewey Trough includes subglacial till, covered with an up to 2.5 m thick drape of glacimarine sediments, the latter reflecting the transitions from sub-ice stream, to glacier proximal and glacier distal deposits.
- The ice stream draining through Store Koldewey Trough probably originated from the area presently covered with the Storstrømmen ice stream. It reached a thickness exceeding the height of the mountains on Store
Koldewey Island and Germania Land, leading to ice flow independent of the subglacial topography during full glacial conditions and an early phase of the deglaciation.
- Grounding zone wedges and various types of recessional moraines (transverse and saw-tooth recessional moraines) within the trough provide evidence that multiple halts and/or readvances interrupted the deglaciation. The formation of the grounding zone wedges probably took at least 130 years. Assuming that
the recessional moraines were annually formed, the distances between the moraines indicate that the grounding line locally retreated between 80 to 400 meters/year during the deglaciation.
- The complex geomorphology in Store Koldewey Trough is attributed to the trough shallowing and narrowing towards the coast, affecting the formation of grounding zone wedges.
- Ice sheet thinning during a late stage of the deglaciation led to topographically controlled ice flow, leading to
diversion of the ice stream to Jøkelbugten and Dove Bugt, and, in consequence, terminating sediment supply to Store Koldewey Trough.

*Author contributions.* The idea of the study developed from repeated discussions among the authors of this study
and the opportunity to participate in multiple scientific cruises to the study area. ILO, JS and TAR collected the new data during the *TUNU VII* cruise. The geophysical and lithological data were interpreted by ILO in collaboration with MF, JSL and TAR. ILO conducted sample preparation and analyses of sediment data with help from KH. ILO wrote the manuscript with contributions from all authors.

*Competing interests*. The authors declare that they have no conflict of interest.

*Acknowledgements* We thank the captains, crews and participants onboard the 2013-, 2015- and 2017-cruises with RV *Helmer Hanssen*, arranged by the TUNU-program led by J.S Christiansen, UiT The Arctic University of Norway. Furthermore, we acknowledge our cruise engineers B. R. Olsen and S. Iversen, as well as our laboratory
engineers T. Dahl, I. Hald and K. Monsen for their technical support during and after the cruises. Finally, we would like to thank Editor Chris Stokes and our two referees, including Sarah Greenwood, for their thorough and critical reviews, as well as for their constructive comments that improved the manuscript.

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

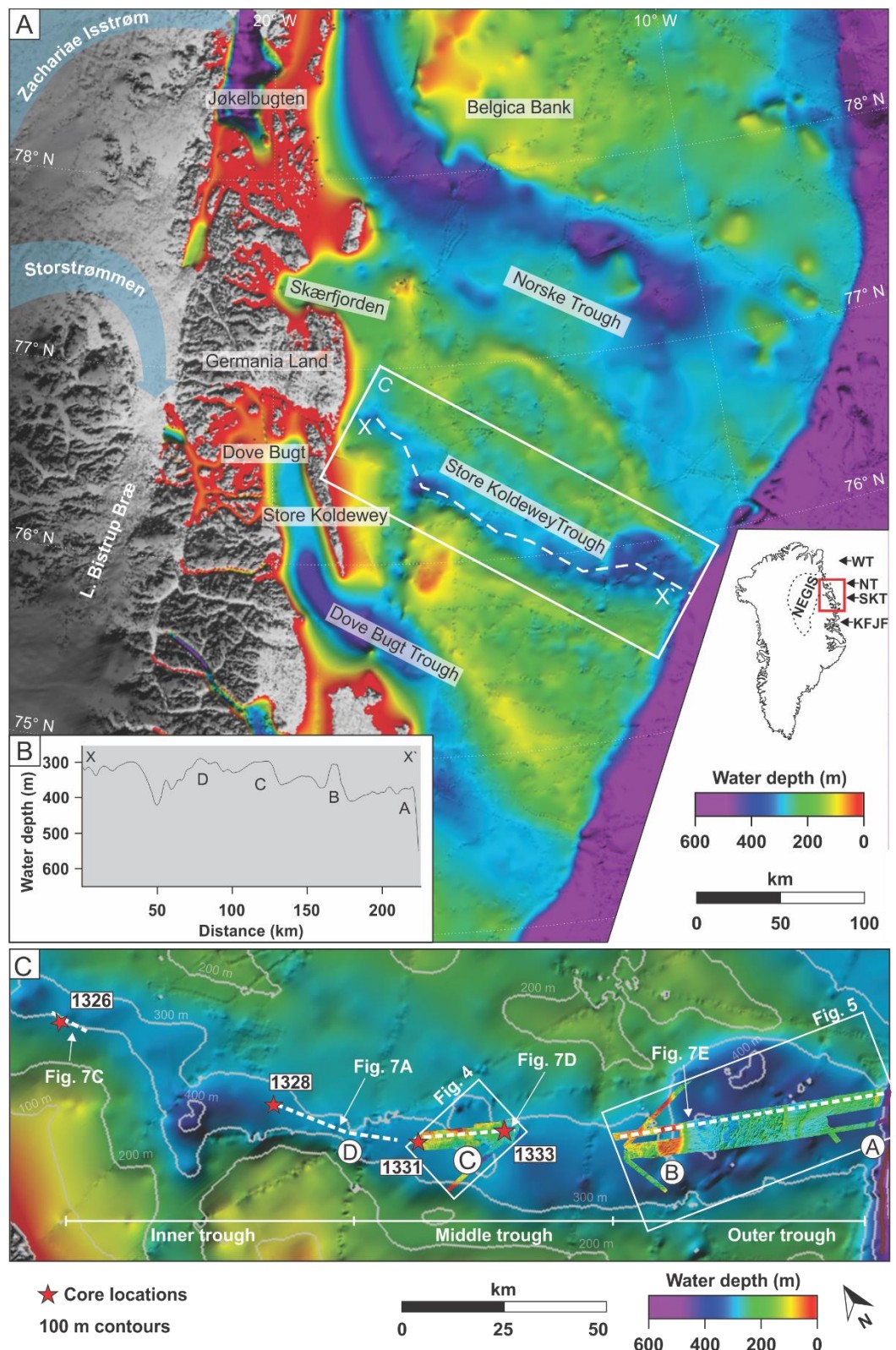

**Figure 1:** (A) Overview of the regional bathymetry and the hinterland topography of NE Greenland (from IBCAO v.4.0; Jakobsson et al., 2020) including geographical names. The small map shows Greenland and the outline of the North East Greenland Ice Stream, with the red box showing the study area detailed in (A). The locations of Westwind Trough (WT), Norske Trough (NT), Store Koldewey Trough (SKT) and Kejser Franz Josef Fjord (KFJF) are indicated. The white dashed line shows the location of bathymetric profile shown in (B). (B) Bathymetric profile along the central axis of Store Koldewey Trough. The labels A-D show the locations of interpreted grounding-zone wedges as described by Laberg et al. (2017). (C) Large-scale bathymetry of Store Koldewey Trough (from IBCAO v.4.0; Jakobsson et al., 2020) including the swath bathymetry data analyzed in this study. The labels A-D represent grounding-zone wedges (adapted from Laberg et al. 2017), red stars show core locations.

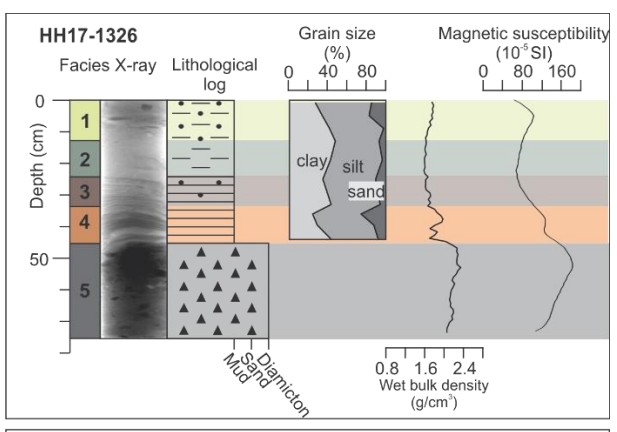

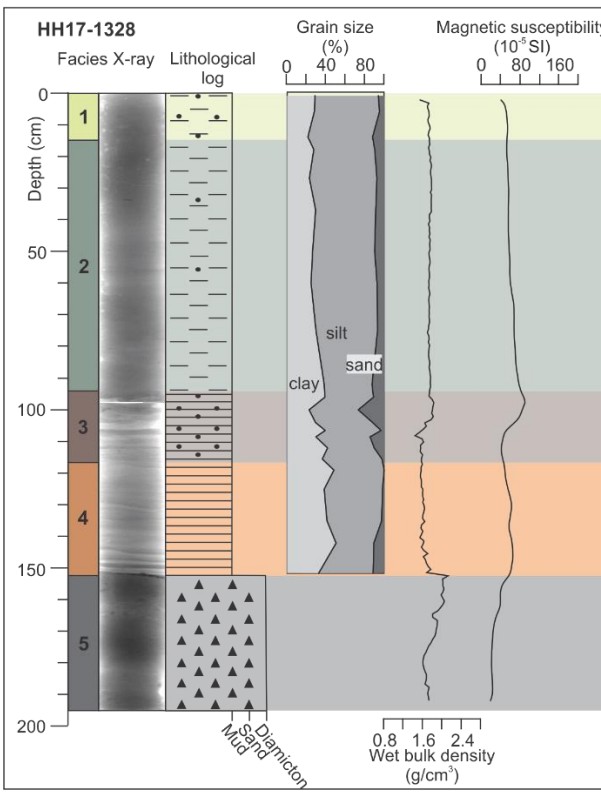

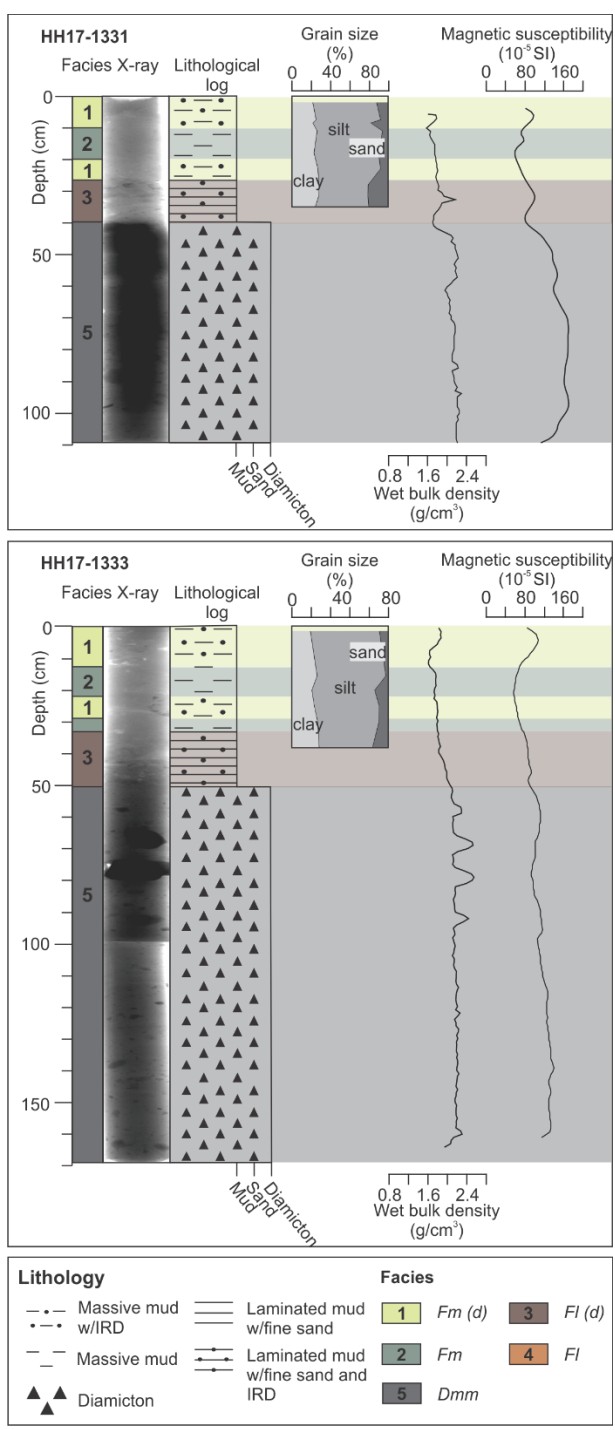

**Figure 2.** Lithofacies logs, X-radiographs, grain-size distribution and physical properties for the cores from inner and middle Store Koldewey Trough. The darker grey tones in the X-radiographs reflect higher density, whereas brighter grey tones reflect lower density. The locations of the sediment cores are shown in Fig. 1C.

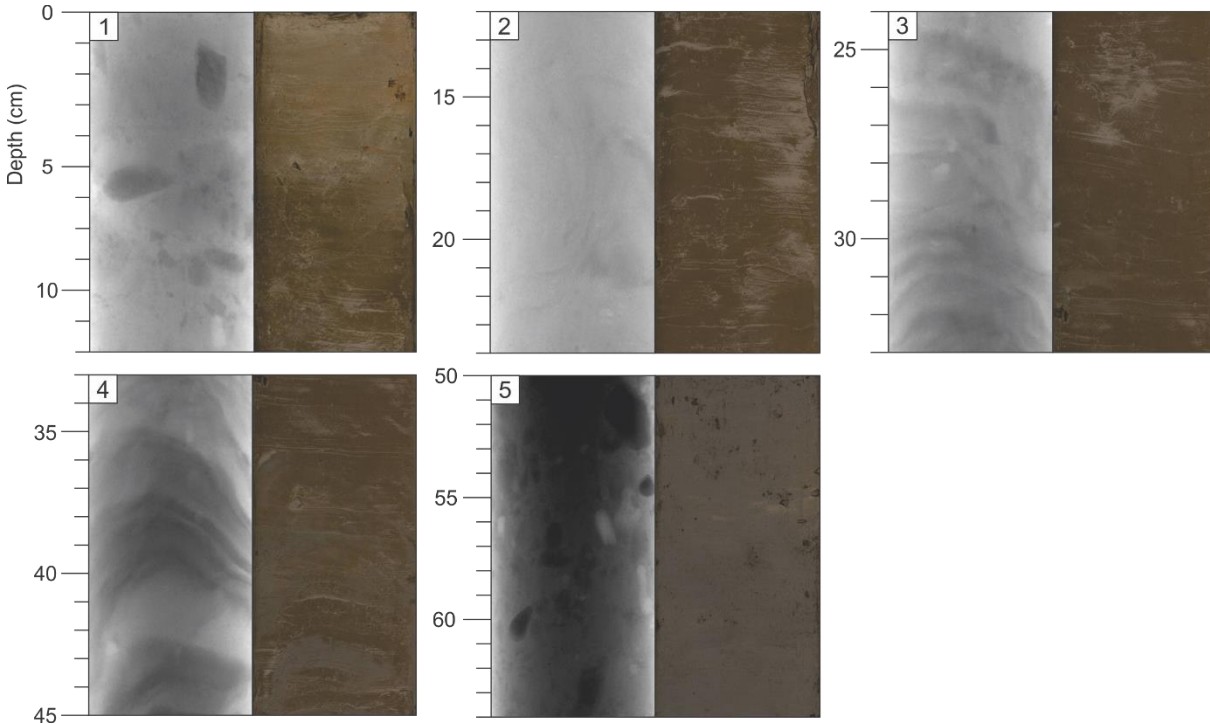

**Figure 3**. X-radiographs with associated photographs of representative lithofacies in this study, all from core HH17-1326. (1) Massive mud with IRD (Fm (d)). (2) Massive mud (Fm). (3) Laminated mud with occasional IRD (Fl (d)). (4) Laminated mud (Fl). (5) Diamicton (Dmm). The darker grey tones on the X-radiographs reflect higher density, whereas brighter grey tones reflect lower density.

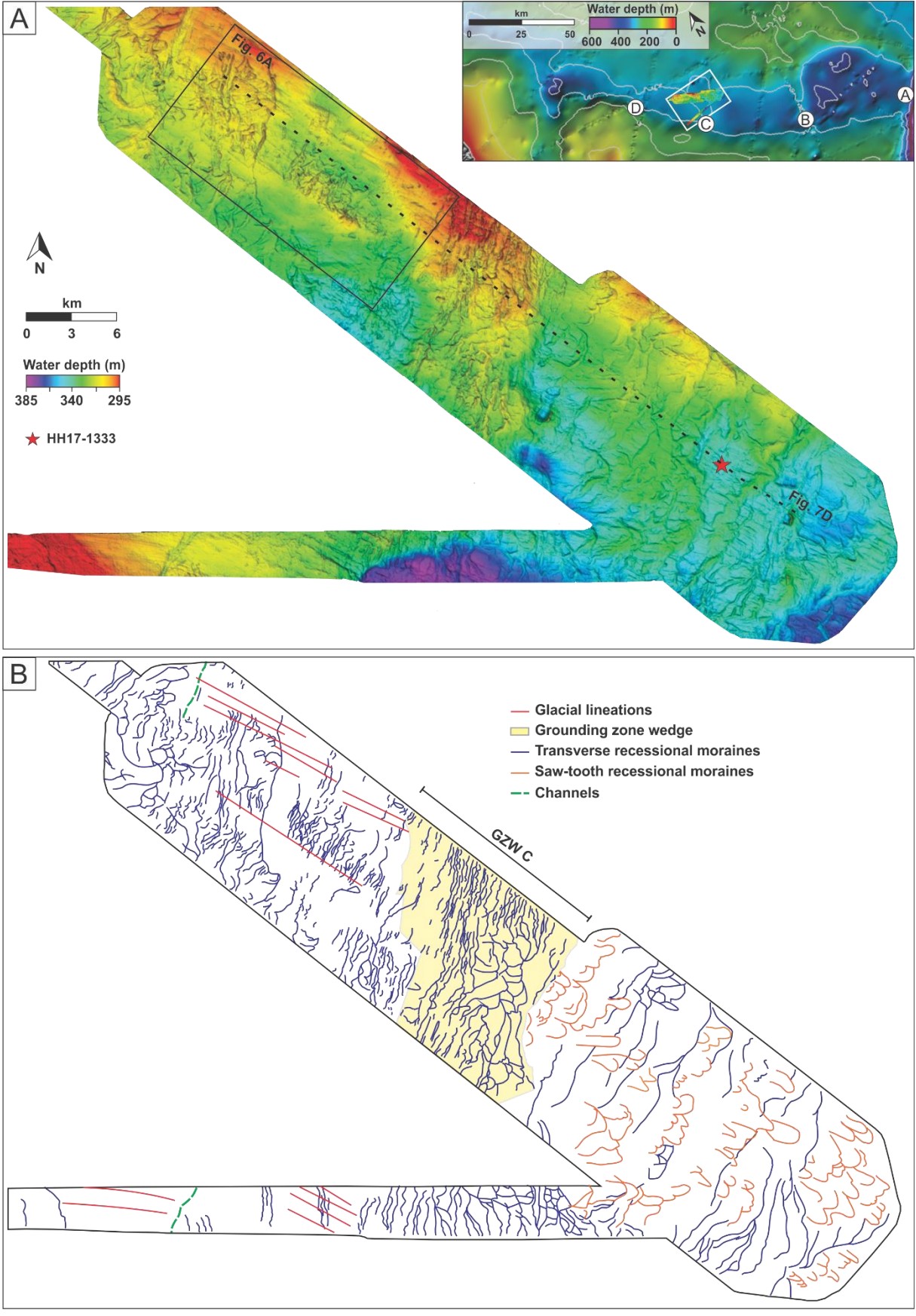

**Figure 4.** (A) Swath bathymetry map from the middle part of Store Koldewey Trough. The locations of grounding zone wedges *A-D* are indicated in inset map (bathymetry from IBCAO v.4.0; Jakobsson et al., 2020). (B) Interpretation and distribution of landforms.

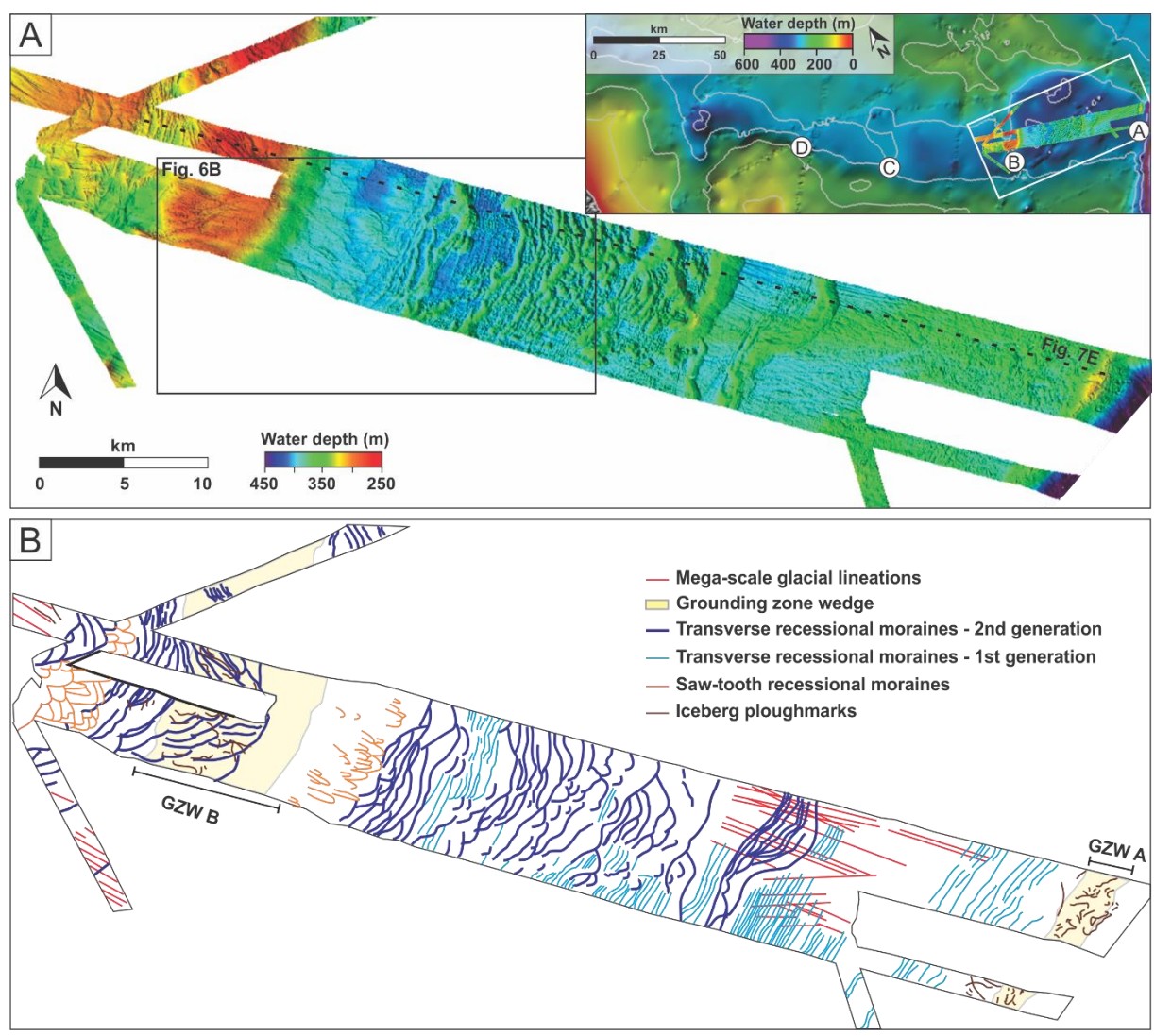

**Figure 5**. (A) Swath bathymetry map from the outer part of Store Koldewey Trough. The locations of grounding zone wedges A-D are indicated in inset map (bathymetry from IBCAO v.4.0; Jakobsson et al., 2020). (B) Interpretation and distribution of landforms modified after Laberg et al. (2017), supplemented with new data.

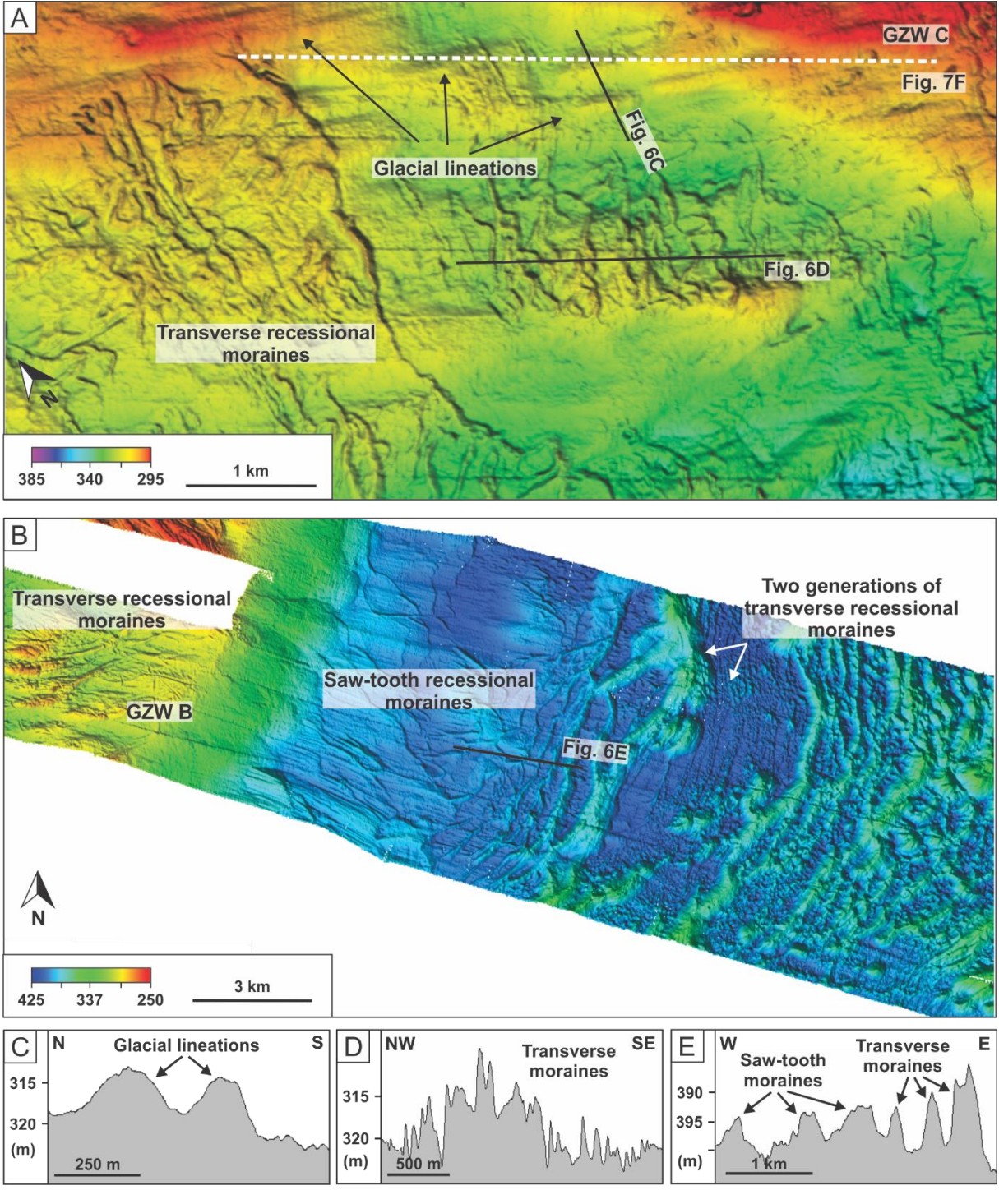

**Figure 6.** (A) Examples of glacial lineations and transverse recessional moraines. (B) Examples of saw-tooth- and transverse recessional moraines. (C) Bathymetric cross-profile of glacial lineations. (D) Bathymetric cross-profile of transverse moraines. (E) Bathymetric profile of saw-tooth moraines and transverse moraines.

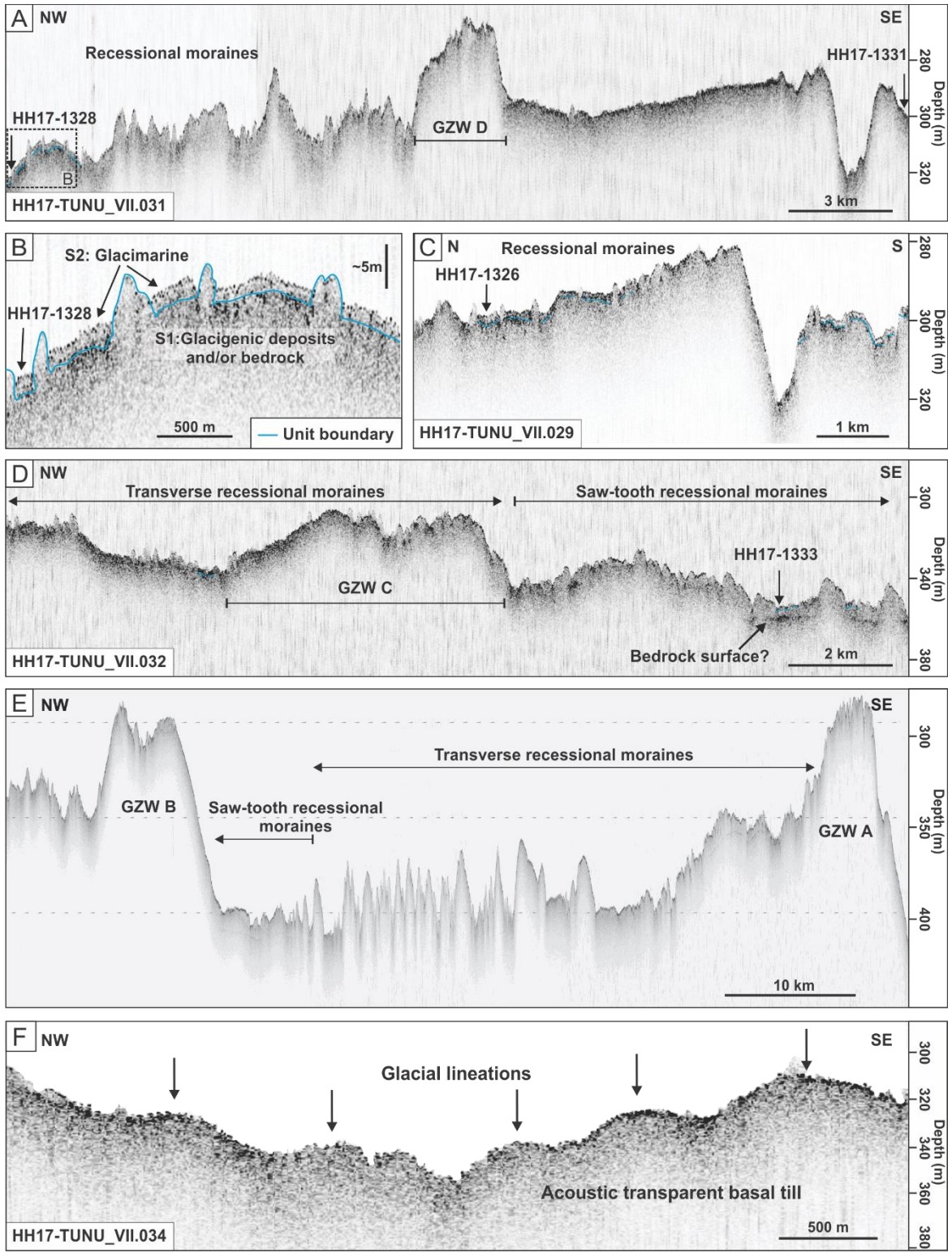

**Figure 7.** High-resolution seismic profiles from Store Koldewey Trough. See Fig. 1C for locations. (A) Chirp line HH17-TUNU_VII.031 across grounding zone wedge D and recessional moraines from the middle trough area. Projected positions of sediment cores HH17-1328 and HH17-1331 are shown. Black dotted rectangle shows extent of the profile in (B). (B) Part of chirp line HH17-TUNU_VII.031 showing a zoom-in example of the configuration of units S1 and S2. (C) Chirp line HH17-TUNU_VII.029 from the inner part of the trough, with recessional moraines. Projected position of sediment core HH17-1326 is indicated. (D) Chirp line HH17-TUNU_VII.032 across grounding zone wedge C. The locations for transverse recessional moraines, saw-tooth recessional moraines and sediment core HH17-1333 are indicated. (E) Chirp line across grounding zone wedge A and B, separated by saw-tooth- and transverse recessional moraines. Modified from Laberg et al. (2017). (F) Part of chirp sub-bottom profile HH17-TUNU_VII.034 showing the acoustically transparent deposits interpreted as basal till/glacial lineations. The ridges of the latter are indicated with arrows.

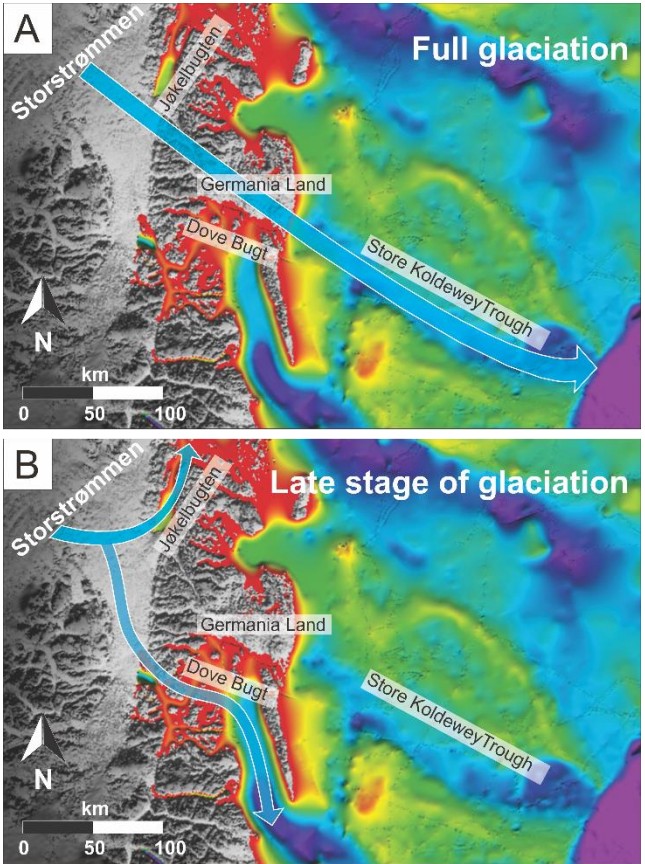

**Figure 8.** Reconstruction of inferred paleo ice-flow directions showing A) paleo ice-flow unrelated to the underlying topography during full glaciation and B) ice drainage paths during a late stage of glaciation. (Bathymetry and topography from IBCAO v.4.0; Jakobsson et al., 2020).

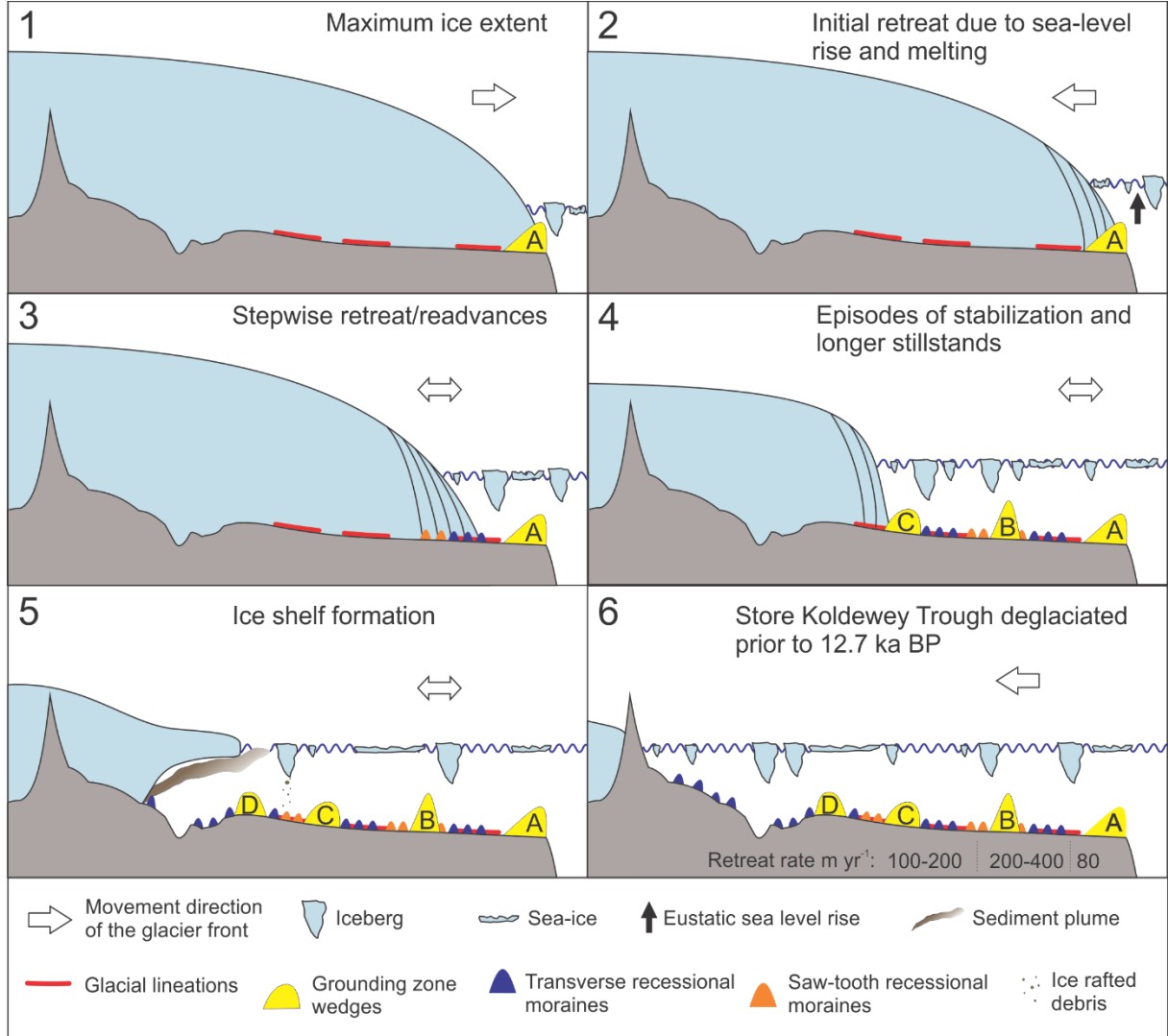

**Figure 9**. Reconstruction of the ice sheet dynamics in Store Koldewey Trough. Stage 1-6 show the maximum ice extent of the ice stream, as well as the ice-stream margin positions during the following deglaciation. Icebergs and sea-ice indicate iceberg calving and ice rafting. The deglaciation age in stage 6 is based on cosmogenic nuclide dating on Store Koldewey Ø (Skov et al., 2020).

| Core ID | Latitude (ºN) | Longitude (ºW) | Water depth (m) | Recovery (cm) |
|---|---|---|---|---|
| HH17-1326-GC-TUNU | 76º21.55´ | 17º05.54´ | 294 | 75 |
| HH17-1328-GC-TUNU | 76º14.07´ | 16º05.42´ | 316 | 195 |
| HH17-1331-GC-TUNU | 76º06.50´ | 15º07.25´ | 306 | 110 |
| HH17-1333-GC-TUNU | 76º00.41´ | 14º09.36´ | 345 | 169 |

**Table 1.** Core locations, water depths and recoveries.

| Lithofacies | 5 - *Dmm* | 4 - *Fl* | 3 - *Fl (d)* | 2 - *Fm* | 1 - *Fm (d)* |
|---|---|---|---|---|---|
| HH17-1326-GC-TUNU (75 cm) | 45 cm - end of core | 33-45 cm | 24-33 cm | 12-24 cm | Top of core - 12 cm |
| HH17-1328-GC-TUNU (195 cm) | 152 cm - end of core | 117-152 cm | 94-117 cm | 14-94 cm | Top of core - 14 cm |
| HH17-1331-GC-TUNU (110 cm) | 40 cm - end of core | Absent | 26-40 cm | 10-20 cm | Top of core - 10 cm 20-26 cm |
| HH17-1333-GC-TUNU (169 cm) | 51 cm - end of core | Absent | 33 cm - 51 cm | 12-22 cm 28-33 cm | Top of core - 12 cm 22-28 cm |
| Lithology | Diamicton, massive and matrix-supported with a sandy mud matrix. Randomly oriented clasts | Laminated mud with fine sandy layers | Laminated mud with fine sandy layers and dropstones | Massive mud with rare dropstones | Massive mud with occasional dropstone |
| Color (Munsell Soil Color Chart) | Very dark gray (2.5Y 3/0) | Dark gray (10YR 4/1) | Dark gray (10YR 4/1) | Olive gray (5Y (/2) Dark gray (10YR 4/1) | Dark grayish brown (2.5Y 4/2) Dark olive gray (5Y 3/2) Olive gray (5Y (4/2) Brown (7.5YR 4/2) |
| Clast amount | High amounts | Absent | Scattered in layers | Rare | Sections containing clasts |
| Bioturbation | Absent | Absent | Absent | Little to moderate | Little |
| Lower unit boundary | Not recovered | Sharp | Gradational or sharp | Gradational | Gradational |
| Upper unit boundary | Sharp | Gradational | Gradational | Gradational | Top of cores |
| Bulk density (g/cm$^3$) | 1.61-2.55 | 1.54-2.03 | 1.60-2.15 | 1.60-1.84 | 1.55-1.78 |
| Magnetic susceptibility (10$^{-5}$ SI) | 20-182 | 46-148 | 66-100 | 53-114 | 41-106 |
| Sedimentary environment | Subglacial till | Proximal glacimarine sedimentation from suspension settling. Sub-ice shelf environment | Proximal glacimarine sedimentation from suspension settling. Ice rafting in a glacimarine calving zone | Distal glacimarine sedimentation dominated by suspension settling. Ice rafting is limited | Distal glacimarine sedimentation dominated by suspension settling. Enhanced ice rafting |

**Table 2.** Overview of the main properties and compositional characteristics of the lithofacies, including depositional environment.

|                                | Length (km) | Width (m)    | Relief (m) | Spacing (m)    |
|--------------------------------|-------------|--------------|------------|----------------|
| Glacial lineations             | 1.5->9      | 150-500      | 4-8        | 200-700        |
| Grounding zone wedges          | N/A         | 3500-10,000  | 35-100     | 45,000-60,000  |
| Transverse recessional moraines| N/A         | <2200        | <50        | 50-500         |
| Saw-tooth recessional moraines | <1.3        | 170-1100     | 5-30       | N/A            |
| Channels                       | N/A         | 150-300      | 3-10       | N/A            |

**Table 3.** Dimensions of submarine landforms.