# Peer review of "Last Glacial ice sheet dynamics offshore NE Greenland – a case study from Store Koldewey Trough"

_The Cryosphere, 2019_

## Referee Comment (RC1) · Anonymous Referee #1 · 14 Feb 2020

**Review: Olsen *et al.***

**Last Glacial ice-sheet dynamics offshore NE Greenland – a case study from Store Koldewey Trough.**

This manuscript presents a mix of new and previously published (by the co-authors) geophysical, geomorphological and sediment core data from the continental shelf off NE Greenland – a region for which we have limited knowledge of ice extent, behaviour or retreat dynamics at and following the last glacial maximum. This sector merits investigation since it is presently drained by the largest ice stream in Greenland whose geometry is unusual and driving mechanism not well understood; it has a broad continental shelf (space to accommodate significant expansion) dissected by troughs (past ice streaming, potentially with a rather different regime to today); and ice-ocean feedbacks through deglaciation are potentially important/variable given the location near the zone of exchange between Atlantic and Arctic waters.

A large part of the geophysical data has been previously published (Laberg et al. 2017), and it should be made more explicit in this manuscript that mapping and interpretations from these data is not new (or should highlight explicitly if earlier interpretations are revised here). The mid-shelf dataset is, as far as I'm aware, new, as are the core data and interpretations.

I would contest some of the landform interpretations (detailed below) and think the sedimentological interpretations could be more specifically discussed with reference to both the authors' analyses and the literature. The Discussion is rather weak and the structure hops around from paragraph to paragraph, without building a sound argument that draws on the evidence presented or rigorously examines the literature. Potentially interesting themes (for example, drivers of retreat; role of melting vs calving; effect of bed slope) therefore aren't fully developed.

There are a handful of grammatical errors in the manuscript (largely subject – verb agreements). Figures are all well put together, but I am not sure that all of figs 8, 9 and 10 (interpretative figures) are required.

*Interpretations*

Sediment cores:

- You refer to grain size and sorting characteristics but present neither for the diamict units (or even diamict matrix) and no sorting data for any unit. Similar for clast count/abundance.
- How consistent is your interpretation of meltwater plumes & underflows with the characteristics of meltwater sediment facies reported elsewhere (eg Witus et al 2014, Smith et al 2017, Prothro et al 2018)?
- The explanations offered for a lack of IRD in an ice-proximal setting (facies 4 vs 3) all assume an ice shelf wouldn't have any basal debris. Examine whether that is valid here.

Landforms:

- This section should make clear that the outer block of multibeam has already been reported on and interpreted by Laberg et al 2017 and isn't new here. I suggest this is acknowledged explicitly, or state that the earlier reported assemblages are re-interpreted here if that is the case (and in which case, why?).
- I question the 'megascale' interpretation of lineations in the mid-trough data. They are rather few, sparse, short and individually distinct compared to a more typically dense ridge-groove arrangement (such as those on the outer shelf shown in the Laberg paper).

- I am not convinced by the examples given of distinct differences between the interpreted recessional moraines, crevasse-squeeze ridges and multi-keel ploughmarks (and the consequent interpretation that they are formed by different mechanisms/in different environments).
  - In Fig 6, I see little difference between the recessional moraines that are slightly irregular (i.e. branch/merge, where part of the grounding line has retreated while pinned elsewhere) and the labelled CSRs. Similarly, the sinuous and (?) composite form of the curvilinear, transverse to ice flow ridges in 6B (interpreted as due to ploughing by icebergs) have the same kind of size and form as curvilinear ridges due to push at the grounding line (i.e. moraines). Most of the supposed ploughmarks in 6B seem to lack an 'inbound' scour that leads to the transverse ridge.
  - All three of these types are distributed throughout the assemblage. Is it not a simpler explanation that moraines are formed by push at the grounding line, and that spatially differential push (small differences in sediment mobility) will create a sinuous and potentially complex product? Three different landform types require fundamentally different ice flow dynamics or environments – how can these be reconciled in this setting?
  - E.g. Ploughing by icebergs requires a fundamentally different environment and time period to grounded, coherent ice approaching the grounding line. What strong evidence is there that these ridges were ploughed in front of (an) iceberg keel(s) rather than pushed up at the grounding line?
  - E.g. Moraines are interpreted here as a product of slow, steady retreat with repeated pauses. Crevasse squeeze ridges, on the other hand, are interpreted as infill of crevasses at the end of a fast flow episode, and the ridges are implicitly synchronously formed rather than in sequence. Yet these two landforms and dynamic interpretations intermingle. At least a discussion of this problem is warranted.
  - If moraines and CSRs are argued to be present here, then mapping them in the same class (same colour) is misleading.
  - Wedge C (? – mid Fig 4 – suggest label wedges A-D where appropriate on Figs 4&5) is superimposed by both (?) moraines and CSRs. Given that wedges are typically associated with prograding debris flows at the grounding line, moraines by local push, and CSRs by basal crevasse infill, how do you reconcile (dynamically) the three being formed on top/immediately adjacent to one another?

Discussion:

- The non-topographically controlled (rather, exceeding topography) aspect of ice stream onset/source is under-developed. What amplitude topography does the ice stream have to override – what ice thickness would ignore a tendency to funnel either side of the higher ground (and is this a reasonable thickness)? Does SKT contrast with other troughs along the coast that are fjord-fed? Could this explain why it might exhibit a different style of retreat to 'typical' troughs? (I note that Laberg et al have already made this interpretation.)
- Regular/many grounding line landforms are interpreted as a product of slow retreat. Why *slow*? Retreat proceeds in steps, yes, but is there independent evidence that these steps occurred slowly? The start of section 5.2 rather treats wedges and moraines (of quite different sizes) as providing the same sort of information: that the ice margin was stable "for a sufficient period" or "had a considerable flux" to build the landforms. I think this passage should explore the basis for "slow" retreat or prolonged standstills, and explain how this model of retreat fits with the later interpretation of surging.
- Arguments for drivers of retreat are muddled.
  - Laberg et al reject the hypothesis of retreat driven by sea level rise, yet here, based on the same data, you favour it. Why? I'm not convinced by the arguments for either (I don't think

you have enough data), but they should be more rigorously discussed. If grounded ice is thick (which you argue for based on it passing over Germania Land) and its lateral extent is curtailed by the continental shelf break rather than because it is supply-limited, then it may be more resilient to sea level rise – the argument here is that a rise should cause ice to go afloat and the grounding line to make a large back-step, but this is contingent on ice thickness being close to the buoyancy threshold, and the bed topography allowing for such a back-step (difficult if landward-shallowing). On page 9, in fact, you point out contrasts between this system with others driven by sea level rise – so why do you call on this mechanism?

- I wouldn't call on evidence for subglacial meltwater flow as the most immediate support of ocean warming-driven retreat (p8 final paragraph) – explain the logic for this.
- A potentially interesting discussion of the roles of meltwater and/or calving doesn't really develop (p9 penultimate paragraph). Are these mutually exclusive modes of retreat? Facies 3 bears similar characteristics to facies 4, except with IRD: do we have continual meltwater with suppressed calving (absence of facies 4)? Or an increase in meltwater-related sediments? The Results report IRD in layers – are you detecting episodic calving events, or continuous delivery? If meltwater sediments and landforms are more abundant towards the inner shelf, is this a temporal effect (i.e. more melt production later in deglaciation), or a spatial effect (e.g. preferential channelisation of water with a certain topography/substrate/ice surface profile)? You ought to be able to develop these ideas more than "retreat style was different" – in what way, and with what significance?

• Comparisons of the numbers/positions of GZWs between troughs with very sparse multibeam/seismic data coverage (e.g. page 9) can at best lead to a speculative conclusion. Contrasting stabilisation points, if that is how a GZW is interpreted, also doesn't necessarily mean asynchronous retreat: retreat may be triggered by a synchronous forcing, but may be locally anchored in different ways. Contrasting pattern doesn't necessarily translate to contrasting timing (or forcing).

• The discussion of surging is under-developed with respect to models for landform formation, drivers and with respect to both literature and the actual data. The basis for the surge interpretation here is the occurrence of crevasse-squeeze ridges. While I'm unconvinced by the figure examples shown, if these are present here then I think the discussion needs to:

- Justify why these must indicate a surge. Is there a difference between a surge (in the sense used here) and a period of ice stream acceleration (externally driven?) which would cause extension and feasibly open up basal crevasses?
- Address the spatial distribution, intermingled with moraines, wedges and iceberg scours – your actual data. Are there multiple patches of CSRs? Do these each, therefore, belong to a different surge? Why? How is this reconciled with "slow and steady" retreat indicated by the moraines? Why should a surge lead to ice front collapse *followed by* increased ice flux? And do you see any evidence for such ice front collapse? This seems at odds with the interpretation of slow, grounded retreat.

*Line-by-line*

P1-line19: "exposed to" increasing ice loss? Rather, 'experienced', or simply 'has increasingly lost mass'…

1-20: 16% of the GIS is…

1-26: instability (also 2-45) – what do you actually mean by this?

1-27: identified as a tipping element

1-30: this sentence is awkward. '…precise predictions of the future potential decay of the GIS…'

P2-first paragraph: you give almost as many references for 'sparse' as 'multiple studies'.

2-15: stepwise – this term isn't especially meaningful, since every pattern showing any kind of paused

grounding line could be said to show 'stepwise' retreat

2-19: *west* of Store Koldewey Trough, *reveals* that…

2-26: I have a bit of a problem with an objective being 'to confirm' something. And in this case, the data you have available with which to 'confirm' (or test) the interpretation of shelf-break glaciation is exactly the same data as has been used to propose it.

3-10: episodic calving

3-12/13: this sentence isn't necessary, unless you make it relevant to your work

3-30: were acquired

4-5: were estimated

4-9: chemical treatment…was conducted

4-35: it could help throughout the presentation of results to use inner/outer shelf or proximal/distal either instead of or as well as compass directions. i.e. here westernmost = innermost, or for example SE of wedge X = distal to or seaward of

5-15: Facies 3 interpretation – laminated mud with sandy layers sounds like Facies 4, interpreted as proximal, so if the 'background' sediments that are interrupted with IRD event layers are the same, does this not imply that the position is sufficiently proximal to still be receiving meltwater sediments?

6-16: grounding zone wedges A-D

6-18: exceeds the…

6-21: wedge A (outermost) has more the shape of a moraine ridge – symmetric form, comparable to the more pronounced of the moraine ridges between here and wedge B. 'A' does not have the asymmetric shape typical of a prograding wedge.

7-13: what do you mean by meltwater runoff from the banks? Proglacial (ie bottom-hugging submarine flows)? Or subglacial from a semi-independent ice sheet sector?

7-29: can you distinguish between a buried bedrock surface and buried till surface (from previous glaciation)?

7-45: This conforms with… (or This is consistent with…)

8-3: covered a minimum length…

8-11: but this is the NE Greenland ice stream (or a distributary of) that you're talking about, so this 'comparison' is a little odd.

8-12: Once fast-flowing ice streams reach…

8-30: what sediment data?

8-39: The occurrence … demonstrates

9-3: ii) trough narrowing

9-second paragraph: this repeats point (i) above

9-19: your high-res data reveal four wedges, but your data coverage is incomplete, so how can you reject the interpretation of six along the whole trough with full (albeit poorer resolution) coverage? You have also consulted IBCAO: do you agree with their interpretations of six wedges, and how does the expression of these in IBCAO compare to their expression in your high res data, where available?

9-fourth paragraph: how does this relate to the rest of your Discussion?

10-5: what sedimentological evidence do you have for winnowing and resuspension?

10-21: grounding zone wedges are not surge-indicative landforms

10-22: is consistent with

*Figures*

1.  Present-day flow directions for Zachariae Isstrøm and Storstrømmen would be useful, and/or

outline of NEGIS. Caption line 2: 'The small map shows…'

4&5. Suggest label wedges A-D on the illustrated mapping in each figure. I also don't think it's helpful or appropriate to show recessional moraines and crevasse-squeeze ridges as part of the same group, since you interpret their formational environment and palaeo-glacial significant differently.

9&10. Both of these figures are not necessary – one or the other should suffice. I'm also not sure all panels of Fig 9 are really necessary – are these really all discrete, distinct 'stages' of retreat that can be clearly defined?

---

## Referee Comment (RC2) · Anonymous Referee #2 · 8 Apr 2020

The manuscript provides a multi-method dataset comprising geophysical, sediment-core and geomorphological data from the little studied area of the NE Greenland continental shelf. Therefore, our understanding of ice sheet history and associated ice-dynamics and sediment processes in this region is poorly constrained. Therefore, a study on this understudied region is welcome and should garner widespread interest. The disappointing aspect of the study was the lack of chronological constraints on the geomorphological dataset and interpretations even though sediment cores were part of the study. Apart from the middle shelf coverage, the swath bathymetry dataset and the interpretation of it seems to be identical to that published in Laberg et al. 2017, but the sediment-core data, middle shelf geophysics and interpretations are new. The

identification of the landforms in swath bathymetric imagery does not appear to be correct. The authors do not make enough use of the sediment core analyses or data, and interpretations need to draw on this data more as well as the literature. The sediment-core aspect of the study could be expanded as core information in NE Greenland is extremely limited in published work to date. The discussion needs to be developed further and there needs to be a natural flow and emergence of a central argument between paragraphs that uses the geomorphological and sedimentological evidence. At times, there does not appear to be a natural link between paragraphs and some paragraphs appear to be dropped in without reference to previous paragraphs.

Section 1-3 Is this paper 'contributing to validation and improvement of numerical models' i.e. will this be examined in this paper based on the data and interpretations presented? If not, then this is a misleading statement and should be altered or removed. I do not see any point in making the observation that "It has been suggested that the northeastern part of the GIS reached the inner or middle parts of the continental shelf during its maximum extent during the last glacial (see Funder et al. 2011 for a review)" as more recent studies of Evans et al. 2009, O Cofaigh et al. 2004, Arndt 2018, Arndt et al. 2015, 2017, Arndt and Evans 2016, and Laberg et al. 2017 show quite clearly that ice went beyond the inner and middle shelf. The authors make this same point so there is no need to repeat an outdated debate. Include Evans et al. 2009, O Cofaigh et al. 2004, Arndt et al. 2015, and Arndt and Evans 2016 in the studies that have indicated ice was much more extensive on the NE Greenland shelf than the original summaries of Funder et al. 1998 and Funder et al. 2011 implied.

The authors need to highlight how the swath bathymetric data presented in this paper differs to that presented in Laberg et al. 2017, and then detail how this study is different to that of Laberg et al. 2017. The same data for the outer shelf is presented again and there needs to be a clear statement or discussion differentiating what is published and what is new. I suggest that the authors add a section detailing what is known about the swath bathymetry and sub-bottom profiler data and implications for ice sheet history

and sedimentary processes of the Laberg et al. 2017 study.

Section 4.1 The range of analyses performed from geochemistry, sediment grain size, shear strength, etc. are outlined in the paper, but there is no reference to the actual data within the description of the lithofacies o, even in the interpretation of the lithofacies or the discussion. For instance, the 'magnetic susceptibility and Ca/Sum ratio vary between each core, with the highest in HH17-1326 and lowest in HH17-1328. Wet bulk density and shear strength are generally high. . .'. This is vague and does not serve the paper well. There is no subsequent use of much of this detailed data when it comes to the discussion of the glacial history later in the paper. I am still uncertain as to the point of including the magnetic susceptibility, XRF and wet bulk density data in this paper beyond including them for the sake of it.

The interpretation of Facies 3 should explain what is meant by 'open conditions' and explain how the 'outer ice-proximal setting' inferred to be the location of the depositional environment differs from that envisaged for Facies 4.

The paper notes the similarity of Facies 2 and 1 apart from the presence of IRD. Does this merely reflect the stochastic behaviour of icebergs rather than anything to do with permanent sea-ice or 'increased influence of drifting ice' in the sense of increased iceberg calving. The differences between the facies is essentially down to the vagaries of iceberg processes.

Section 4.2 I am not convinced that there are MSGL in Figure 4 and 6. The features shown in Figure 5 appear to be lineations rather than MSGL and the description of them only refers to their length as >1.5 km. Do sub-bottom profiler records across the GZWs exist in order to rule out that they are bedrock sills?

Figures 4 and 5 are misleading as the recessional moraines and crevasse squeeze ridges are merged and have the same colour scheme, and it is difficult to distinguish where the crevasse-squeeze ridges are located.
I am not convinced that some of the ridges represent a rhombohedral network indicative of crevasse-squeeze ridges. There appears to be little difference between the recession moraines and the crevasse-squeeze ridges apart from slight differences in morphology that might be linked to variations in grounding line processes and behaviour. The CSR appear to have a limited distribution and are not pervasive or widespread implying that the interpretation of 'surging' is unlikely and that they are more likely to be a localised feature maybe related to complex pattern of recessional moraines linked to ice-margin processes during standstill and retreat. Therefore, the idea of surging behaviour may not be correct and that the landform assemblages only record variable rates of grounding ice margin retreat and stabilisation.

If indeed these features are CSR, why do they have to be associated with a surge rather than an advance/acceleration of an ice stream (linked to mass balance) and formation of basal creavsses due to tensile stress and ice break-up as it steps back to a stillstand position? Also, if it's a surge or even a simple readvance/acceleration of an ice stream, why aren't these features more widespread across the trough floor as presumably, a wider area would stagnate? The limited distribution implies a more complex recessional moraine pattern linked to complex ice retreat in some areas.

I'm not convinced that the features identified as multi-keel iceberg ploughmarks is correct as they appear identical to the recessional moraines in Figure 4, 5 or 6. How would you even differentiate between a multi-keeled iceberg ploughmarks and the intervening ridges they create from those that are recessional moraines?

Section 5 The authors state that "We propose that the Store Koldewey Trough was filled by grounded ice originating from the area presently covered with the Storstrømmen ice stream (Fig. 8A). This implies that the northeastern sector of the GIS reached a thickness allowing the ice stream to flow unrelated to the underlying topography, including the mountain ranges between present day Storstrømmen and Germania Land." This is speculative statement on its own. On what basis or geomorphological evidence are you making this assertion? Why wouldn't Storstrommen have preferentially flowed

along and filled Dove Bugt Trough? The authors then go on to note that "An alternative interpretation is that Store Koldewey Trough had a much smaller drainage-basin, limited to Germania Land (Arndt et al., 2015). However, based on our data, including the observations of mega-scale glacial lineations, recessional moraines and grounding zone wedges, we favor the interpretation of Storstrømmen filling Store Koldewey Trough during full glacial conditions based on the volume of ice needed to fill a trough of this magnitude. We propose that the ice sheet thinned and that the underlying topography controlled the direction of ice flow during a late phase of the last glacial, i.e. that the ice flow from the interior of the GIS was directed to Jøkelbugten in the north and Dove Bugt in the south (Fig. 8B)." What is being proposed is speculative. Therefore, the discussion on the topographic and non-topographic controls on ice stream flow pathways, source and development from one to the other needs to be developed further.

Why does the retreat of the grounded ice margin have to be 'slow' between stillstands? What evidence is used to support this assertion? There are no radiocarbon dates from the study cores that constrain ice stream retreat, so it is not possible to conclude the relative rate of retreat. Evidence from Antarctica shows that ice streams can abandon their groundling zone very quickly and then retreat at variable rates to the next stabilisation point. It is worth exploring the issue of terrain factors (e.g. trough dimensions, trough depth distribution, underlying bed slope, etc.) modulating externally driven ice sheet retreat. The authors should consider the literature on GZW morphology and volume as an indicator of the relative length of time that the grounding line remains stable in one place (e.g. Dowdeswell et al. and Batchelor et al.). The authors need to develop the discussion in terms of what the smaller recessional moraines versus the larger GZWs mean for ice stream retreat rates, length of time of stabilisation and ice margin behaviour during temporary stillstands. For instance, the smaller moraines may be winter advances during stillstand

The authors note that "We interpret the break-up and retreat of the GIS to have happened in two stages; initial retreat by breaking up and calving of grounded ice due to eustatic sea level rise caused by melting of ice at lower latitudes (Lambeck et al., 2014) (Fig. 9: Stage 2) and a second phase of melting driven by ocean warming, possibly due to the onset of inflow of intermediate water masses. The latter is supported by the occurrence of meltwater-channels and laminated sediments interpreted to be a result of excessive meltwater production in the middle and inner parts of the trough". On what basis, evidence or studies are you making this assertion for this region of Greenland, particularly the impact of sea level rise or inflow of intermediate water masses? What intermediate water masses are you referring to? There is no sediment evidence such as iceberg rafted lithofacies recorded in the cores to support iceberg calving and margin retreat due to sea level rise. Meltwater derived sediment facies cannot be used the defining piece of evidence indicating ocean warming retreat as the ice sheet will always produce and discharge meltwater due to the simple fact the ice at the subglacial bed is at pressure melt point. In fact, meltwater sediments will be deposited even when sea levels are rising and causing the ice margin to retreat.

The impact of these external factors will depend on the relative balance between atmospheric warming, precipitation, ocean warming and sea level rise, but stating that sea level rise causes a first stage of retreat is too simplified. For instance, studies in Antarctica show that maximum grounded ice extent in some sectors of the Ross Sea occurred during deglaciation even though there was atmospheric and oceanic warming and sea level rise because precipitation had a more dominate impact on mass balance, but eventually ocean factors dominated to cause retreat. How do you know there are two phases to the retreat of the ice sheet in this region without age constraints? In fact, the geomorphology implies more than two stages to ice sheet retreat. It is also worth noting that ice sheet retreat history is not merely a simple function of sea level rise, ocean warming and atmospheric warming but also due to terrain factors that can modulate ice sheet response and the rate of response to these external factors. The authors note the importance of the terrain for ice sheet retreat but do not really consider the literature that have looked at the impact of terrain factors on ice stream retreat. For

example, Stewart et al. 2012 and Livingstone et al. 2012. These studies show the importance of trough width and depth on the rate of ice sheet retreat and try to quantify rates of retreat.

The authors note that "Based on the varying numbers of GZW's we suggest that retreat/readvances of the ice streams offshore NE Greenland occurred asynchronously." Whilst I agree that it is possible that ice streams over such a large region as NE Greenland will experience asynchronous behaviour, I am not convinced of the evidence that is presented for this assertion. The data from Norske Trough, Westwind Trough and elsewhere do not provide a complete coverage of the respective areas and it is possible that there may be GZWs that exist, but have yet to be discovered undermining the suggestion that the number of GZWs indicates asynchronous ice stream behaviour. Secondly, without chronological constraints on regional ice stream behaviour during deglaciation or the ages of GZWs then the assertion of asynchronous behaviour is speculative.

The authors note that "The present sub-glacial topography of Storstrømmen consists of a reversed bed slope, accompanied by a floating ice tongue (Hill et al., 2018). Thus, a potential future response to increased ocean warming could result in episodes of rapid retreat as the ice front undergoes thinning and/or ice tongue collapse. Such episodes are believed to cause a dynamic response up-glacier, resulting in an accelerated ice flow, contributing directly to sea level rise (Hill et al., 2018)". It is not entirely clear how this statement links, and is relevant, to the previous paragraphs discussing ice sheet behaviour during deglaciation.

The authors equate 'surge' behaviour with an ice stream. Why does the ice stream have to surge rather than simply readvance/accelerate? I am not convinced that the features they describe are crevasse-squeeze ridges but if they are then the section needs to be developed further to explain and justify why the ice stream surges as opposed to accelerate and readvance. The authors also need to explain why the CSR are limited in their spatial extent and distribution within the swath bathymetry dataset

and why they have a close association with the GZW and recessional moraines.

---

## Editor Comment (EC1) · Chris R. Stokes (Editor) · 17 Apr 2020

I'd like to record my thanks to both reviewers for their insightful and constructive reviews of this manuscript. It is clear that whilst both reviewers see merit and potential in the work, they both recommend major revisions to the manuscript before it can be published. I share many of their concerns, particularly the rather awkward relationship between this work and the paper published in Laberg et al. (2017) and the broader implications and significance of the work and some of the interpretations. That said, I am open to a re-submission of a substantially revised and improved manuscript, which would then go back out to review.

Best wishes,

Chris Stokes

---

## Author Comment (AC1) · 24 Jun 2020

*The authors' replies on the comments are written in red, bold and italics*

**Review: Olsen *et al.***

**Last Glacial ice-sheet dynamics offshore NE Greenland – a case study from Store Koldewey Trough.**

This manuscript presents a mix of new and previously published (by the co-authors) geophysical, geomorphological and sediment core data from the continental shelf off NE Greenland – a region for which we have limited knowledge of ice extent, behaviour or retreat dynamics at and following the last glacial maximum. This sector merits investigation since it is presently drained by the largest ice stream in Greenland whose geometry is unusual and driving mechanism not well understood; it has a broad continental shelf (space to accommodate significant expansion) dissected by troughs (past ice streaming, potentially with a rather different regime to today); and ice-ocean feedbacks through deglaciation are potentially important/variable given the location near the zone of exchange between Atlantic and Arctic waters.

A large part of the geophysical data has been previously published (Laberg et al. 2017), and it should be made more explicit in this manuscript that mapping and interpretations from these data is not new (or should highlight explicitly if earlier interpretations are revised here). The mid-shelf dataset is, as far as I'm aware, new, as are the core data and interpretations.

I would contest some of the landform interpretations (detailed below) and think the sedimentological interpretations could be more specifically discussed with reference to both the authors' analyses and the literature. The Discussion is rather weak and the structure hops around from paragraph to paragraph, without building a sound argument that draws on the evidence presented or rigorously examines the literature. Potentially interesting themes (for example, drivers of retreat; role of melting vs calving; effect of bed slope) therefore aren't fully developed.

There are a handful of grammatical errors in the manuscript (largely subject – verb agreements). Figures are all well put together, but I am not sure that all of figs 8, 9 and 10 (interpretative figures) are required.

***We included the data set from Laberg et al. (2017) into this manuscript with the purpose of improving the regional understanding of the bathymetry of Store Koldewey Trough. We have clarified what part of the data that is previously published by Laberg et al. (2017). As part of the inclusion, we have re-interpreted the published data from the outer shelf in greater detail.***

***We have restructured and rewritten the discussion chapter, by adding new paragraphs as well as reorganizing existing paragraphs. New paragraphs focus on possible drivers of retreat, the role local trough topography may have had on the retreating ice front, as well as calculations on GZW volumes and the relative length of time of grounding line stabilization.***

***We removed figure 10, a schematic landform-assemblage model for Store Koldewey Trough.***

*Interpretations*

Sediment cores:

- You refer to grain size and sorting characteristics but present neither for the diamict units (or even diamict matrix) and no sorting data for any unit. Similar for clast count/abundance. ***When mentioning clast amount we refer to the relative abundance, based on visual observations in the X-ray images. We refrained from presenting clast counts as we regarded this irrelevant in the context of the current paper, as it lacks absolute chronologies.***

- How consistent is your interpretation of meltwater plumes & underflows with the characteristics of meltwater sediment facies reported elsewhere (eg Witus et al 2014, Smith et al 2017, Prothro et al 2018)? ***We think that our interpretation is consistent with the mentioned publications, given the data available.***

- The explanations offered for a lack of IRD in an ice-proximal setting (facies 4 vs 3) all assume an ice

shelf wouldn't have any basal debris. Examine whether that is valid here. *We have not excluded the presence of IRD in an ice shelf. However, we have provided possible explanations for an absence of IRD in such a setting and moved the previous suggestions in chapter 4.1.2 (Facies 4) to the discussion chapter (chapter 5.2 Glacial dynamics during deglaciation).*

Landforms:

- This section should make clear that the outer block of multibeam has already been reported on and interpreted by Laberg et al 2017 and isn't new here. I suggest this is acknowledged explicitly, or state that the earlier reported assemblages are re-interpreted here if that is the case (and in which case, why?). *In the revised version of the manuscript we mention that the data set from the outer shelf has been published by Laberg et al. (2017), however, we have re-interpreted the data set in greater detail.*

- I question the 'megascale' interpretation of lineations in the mid-trough data. They are rather few, sparse, short and individually distinct compared to a more typically dense ridge-groove arrangement (such as those on the outer shelf shown in the Laberg paper). *We understand the reviewer`s point. However, we keep our suggestion that the landforms are fragments of/partly buried MSGLs, because the lengths/width ratios exceed 10:1 (cf. Clark, 1993).*

- I am not convinced by the examples given of distinct differences between the interpreted recessional moraines, crevasse-squeeze ridges and multi-keel ploughmarks (and the consequent interpretation that they are formed by different mechanisms/in different environments).
  - In Fig 6, I see little difference between the recessional moraines that are slightly irregular (i.e. branch/merge, where part of the grounding line has retreated while pinned elsewhere) and the labelled CSRs. Similarly, the sinuous and (?) composite form of the curvilinear, transverse to ice flow ridges in 6B (interpreted as due to ploughing by icebergs) have the same kind of size and form as curvilinear ridges due to push at the grounding line (i.e. moraines). Most of the supposed ploughmarks in 6B seem to lack an 'inbound' scour that leads to the transverse ridge.
  - All three of these types are distributed throughout the assemblage. Is it not a simpler explanation that moraines are formed by push at the grounding line, and that spatially differential push (small differences in sediment mobility) will create a sinuous and potentially complex product? Three different landform types require fundamentally different ice flow dynamics or environments – how can these be reconciled in this setting?
  - E.g. Ploughing by icebergs requires a fundamentally different environment and time period to grounded, coherent ice approaching the grounding line. What strong evidence is there that these ridges were ploughed in front of (an) iceberg keel(s) rather than pushed up at the grounding line?
  - E.g. Moraines are interpreted here as a product of slow, steady retreat with repeated pauses. Crevasse squeeze ridges, on the other hand, are interpreted as infill of crevasses at the end of a fast flow episode, and the ridges are implicitly synchronously formed rather than in sequence. Yet these two landforms and dynamic interpretations intermingle. At least a discussion of this problem is warranted.
  - If moraines and CSRs are argued to be present here, then mapping them in the same class (same colour) is misleading.
  - Wedge C (? – mid Fig 4 – suggest label wedges A-D where appropriate on Figs 4&5) is superimposed by both (?) moraines and CSRs. Given that wedges are typically associated with prograding debris flows at the grounding line, moraines by local push, and CSRs by basal crevasse infill, how do you reconcile (dynamically) the three being formed on

top/immediately adjacent to one another?

*We appreciate the extensive comment of the referee! Based on that, we revisited the data set and changed our interpretations from crevasse-squeeze ridges and multi-keel ploughmarks to saw-tooth moraines. Therefore, we rewrote the part of the result chapter regarding these specific landforms as well as the following discussion chapter.*

Discussion:

- The non-topographically controlled (rather, exceeding topography) aspect of ice stream onset/source is under-developed. What amplitude topography does the ice stream have to override – what ice thickness would ignore a tendency to funnel either side of the higher ground (and is this a reasonable thickness)? Does SKT contrast with other troughs along the coast that are fjord-fed? Could this explain why it might exhibit a different style of retreat to 'typical' troughs? (I note that Laberg et al have already made this interpretation.) *We have elaborated on this topic by providing information on the altitude that the Storstrømmen Ice Stream had to overcome to drain into Store Koldewey Trough (single peaks of 500-900 m), complimented with modelling results of paleo-ice sheet thickness on Germania Land during LGM (1000-1500 m; Fleming and Lambeck (2004) and Heinemann et al. (2014)).*

- Regular/many grounding line landforms are interpreted as a product of slow retreat. Why *slow*? Retreat proceeds in steps, yes, but is there independent evidence that these steps occurred slowly? The start of section 5.2 rather treats wedges and moraines (of quite different sizes) as providing the same sort of information: that the ice margin was stable "for a sufficient period" or "had a considerable flux" to build the landforms. I think this passage should explore the basis for "slow" retreat or prolonged standstills, and explain how this model of retreat fits with the later interpretation of surging. *The terms "slow" and "episodic" retreat have been introduced by both Ó Cofaigh et al. (2008) and Dowdeswell et al. (2008) discussing styles of ice retreat accompanied with the formation of recessional moraines and grounding zone wedges, respectively. We wish to continue using these terms and have, therefore, rephrased the paragraph, hopefully making our use of terms more clear to the reader.*

- Arguments for drivers of retreat are muddled.
  - Laberg et al reject the hypothesis of retreat driven by sea level rise, yet here, based on the same data, you favour it. Why? I'm not convinced by the arguments for either (I don't think you have enough data), but they should be more rigorously discussed. If grounded ice is thick (which you argue for based on it passing over Germania Land) and its lateral extent is curtailed by the continental shelf break rather than because it is supply-limited, then it may be more resilient to sea level rise – the argument here is that a rise should cause ice to go afloat and the grounding line to make a large back-step, but this is contingent on ice thickness being close to the buoyancy threshold, and the bed topography allowing for such a back-step (difficult if landward-shallowing). On page 9, in fact, you point out contrasts between this system with others driven by sea level rise – so why do you call on this mechanism?
  - I wouldn't call on evidence for subglacial meltwater flow as the most immediate support of ocean warming-driven retreat (p8 final paragraph) – explain the logic for this.
  - A potentially interesting discussion of the roles of meltwater and/or calving doesn't really develop (p9 penultimate paragraph). Are these mutually exclusive modes of retreat? Facies 3 bears similar characteristics to facies 4, except with IRD: do we have continual meltwater with suppressed calving (absence of facies 4)? Or an increase in meltwater-related sediments? The Results report IRD in layers – are you detecting episodic calving events, or continuous delivery? If meltwater sediments and landforms are more abundant towards the inner shelf,

is this a temporal effect (i.e. more melt production later in deglaciation), or a spatial effect (e.g. preferential channelisation of water with a certain topography/substrate/ice surface profile)? You ought to be able to develop these ideas more than "retreat style was different" – in what way, and with what significance? *We have rewritten this part of the manuscript, providing a more in-depth discussion of the relationship between local trough geometry and locations of the GZWs, as well as the sedimentary environments regarding the different lithofacies.*

- Comparisons of the numbers/positions of GZWs between troughs with very sparse multibeam/seismic data coverage (e.g. page 9) can at best lead to a speculative conclusion. Contrasting stabilisation points, if that is how a GZW is interpreted, also doesn't necessarily mean asynchronous retreat: retreat may be triggered by a synchronous forcing, but may be locally anchored in different ways. Contrasting pattern doesn't necessarily translate to contrasting timing (or forcing). *We agree that there might be undiscovered GZWs in Norske Trough and Westwind Trough. We rephrased this paragraph, focusing on the presentations of facts, rather than speculating on differences in deglaciation dynamics between different troughs.*

- The discussion of surging is under-developed with respect to models for landform formation, drivers and with respect to both literature and the actual data. The basis for the surge interpretation here is the occurrence of crevasse-squeeze ridges. While I'm unconvinced by the figure examples shown, if these are present here then I think the discussion needs to:
  - Justify why these must indicate a surge. Is there a difference between a surge (in the sense used here) and a period of ice stream acceleration (externally driven?) which would cause extension and feasibly open up basal crevasses?
  - Address the spatial distribution, intermingled with moraines, wedges and iceberg scours – your actual data. Are there multiple patches of CSRs? Do these each, therefore, belong to a different surge? Why? How is this reconciled with "slow and steady" retreat indicated by the moraines? Why should a surge lead to ice front collapse *followed by* increased ice flux? And do you see any evidence for such ice front collapse? This seems at odds with the interpretation of slow, grounded retreat. *We have re-interpreted the landforms initially suggested to be related to surging, to be saw-tooth moraines. Thus, the discussion of surges is irrelevant for the manuscript, so we have removed this section.*

*Line-by-line*
P1-line19: "exposed to" increasing ice loss? Rather, 'experienced', or simply 'has increasingly lost mass'… *Corrected.*
1-20: 16% of the GIS is… *Corrected.*
1-26: instability (also 2-45) – what do you actually mean by this? *Here we use the term ´instability´ to refer to a possible disequilibrium within the GIS caused by external forces.*
1-27: identified as a tipping element *Corrected.*
1-30: this sentence is awkward. '…precise predictions of the future potential decay of the GIS…' *Changed.*
P2-first paragraph: you give almost as many references for 'sparse' as 'multiple studies'. *We have added more examples of references.*
2-15: stepwise – this term isn't especially meaningful, since every pattern showing any kind of paused grounding line could be said to show 'stepwise' retreat. *We use the term ´stepwise retreat´ to explain interruptions in the retreat. This term has been used in other articles and we therefore wish to keep it.*
2-19: *west* of Store Koldewey Trough, *reveals* that… *Corrected.*
2-26: I have a bit of a problem with an objective being 'to confirm' something. And in this case, the data you have available with which to 'confirm' (or test) the interpretation of shelf-break glaciation is exactly the same data as has been used to propose it. *Changed.*

3-10: episodic calving *Corrected.*

3-12/13: this sentence isn't necessary, unless you make it relevant to your work *We agree and have removed this sentence.*

3-30: were acquired *Corrected.*

4-5: were estimated *Corrected.*

4-9: chemical treatment…was conducted *Corrected.*

4-35: it could help throughout the presentation of results to use inner/outer shelf or proximal/distal either instead of or as well as compass directions. i.e. here westernmost = innermost, or for example SE of wedge X = distal to or seaward of *We have rephrased from compass directions where practical to make it easier for the reader to follow.*

5-15: Facies 3 interpretation – laminated mud with sandy layers sounds like Facies 4, interpreted as proximal, so if the 'background' sediments that are interrupted with IRD event layers are the same, does this not imply that the position is sufficiently proximal to still be receiving meltwater sediments? *We have rephrased our interpretation, making it more clear that both suspension settling and iceberg rating is present.*

6-16: grounding zone wedges A-D *Corrected.*

6-18: exceeds the… *Corrected.*

6-21: wedge A (outermost) has more the shape of a moraine ridge – symmetric form, comparable to the more pronounced of the moraine ridges between here and wedge B. 'A' does not have the asymmetric shape typical of a prograding wedge. *Noted. We base our interpretation on the fact that other ridges with similar dimensions and locations on the continental shelf of Greenland are interpreted as GZWs.*

7-13: what do you mean by meltwater runoff from the banks? Proglacial (ie bottom-hugging submarine flows)? Or subglacial from a semi-independent ice sheet sector? *We believe the channels are not important for reconstruction the ice dynamics and have therefore made a simple interpretation of their origin, suggesting that they are formed during deglaciation and are related to meltwater. In order to further study their genesis, additional data is needed.*

7-29: can you distinguish between a buried bedrock surface and buried till surface (from previous glaciation)? *We have included Petersen et al. (2015), showing that there is a thick Paleogene sedimentary succession offshore NE Greenland, ruling out bedrock sills.*

7-45: This conforms with… (or This is consistent with…) *Corrected.*

8-3: covered a minimum length… *Corrected.*

8-11: but this is the NE Greenland ice stream (or a distributary of) that you're talking about, so this 'comparison' is a little odd. *We find it interesting that a similar flow feature is identified in modern day NEGIS.*

8-12: Once fast-flowing ice streams reach… *Corrected.*

8-30: what sediment data? *We have clarified that Stein et al. (1996) presented terrigenous, coarse grained material along the continental slope off NE Greenland.*

8-39: The occurrence … demonstrates *Corrected.*

9-3: ii) trough narrowing *Corrected.*

9-second paragraph: this repeats point (i) above *Corrected.*

9-19: your high-res data reveal four wedges, but your data coverage is incomplete, so how can you reject the interpretation of six along the whole trough with full (albeit poorer resolution) coverage? You have also consulted IBCAO: do you agree with their interpretations of six wedges, and how does the expression of these in IBCAO compare to their expression in your high res data, where available? *The paragraph has been altered and include now the possibility for GZWs outside our data coverage.*

9-fourth paragraph: how does this relate to the rest of your Discussion? *We see your point and have removed this paragraph.*

10-5: what sedimentological evidence do you have for winnowing and resuspension? *We have rephrased this statement.*

10-21: grounding zone wedges are not surge-indicative landforms *Corrected.*

10-22: is consistent with *Corrected.*

*Figures*

1. Present-day flow directions for Zachariae Isstrøm and Storstrømmen would be useful, and/or outline of NEGIS. *Included.*

   Caption line 2: 'The small map shows…' *Corrected.*

4&5. Suggest label wedges A-D on the illustrated mapping in each figure. I also don't think it's helpful or appropriate to show recessional moraines and crevasse-squeeze ridges as part of the same group, since you interpret their formational environment and palaeo-glacial significant differently. *GZW A-D have been labeled, whilst the mapping of landforms have been updated.*

9&10. Both of these figures are not necessary – one or the other should suffice. I'm also not sure all panels of Fig 9 are really necessary – are these really all discrete, distinct 'stages' of retreat that can be clearly defined? *We agree and have therefore simplified figure 9. Figure 10 has been removed from the manuscript.*

---

## Author Comment (AC2) · 24 Jun 2020

The manuscript provides a multi-method dataset comprising geophysical, sedimentcore and geomorphological data from the little studied area of the NE Greenland continental shelf. Therefore, our understanding of ice sheet history and associated icedynamics and sediment processes in this region is poorly constrained. Therefore, a study on this understudied region is welcome and should garner widespread interest. The disappointing aspect of the study was the lack of chronological constraints on the geomorphological dataset and interpretations even though sediment cores were part of the study. Apart from the middle shelf coverage, the swath bathymetry dataset and the interpretation of it seems to be identical to that published in Laberg et al. 2017, but the sediment-core data, middle shelf geophysics and interpretations are new. The identification of the landforms in swath bathymetric imagery does not appear to be correct. The authors do not make enough use of the sediment core analyses or data, and interpretations need to draw on this data more as well as the literature. The sedimentcore aspect of the study could be expanded as core information in NE Greenland is extremely limited in published work to date. The discussion needs to be developed further and there needs to be a natural flow and emergence of a central argument between paragraphs that uses the geomorphological and sedimentological evidence. At times, there does not appear to be a natural link between paragraphs and some paragraphs appear to be dropped in without reference to previous paragraphs. ***We acknowledge the feedback from the referee and tried to address the issues mentioned in our revision. Please see below for details.***

Section 1-3 Is this paper 'contributing to validation and improvement of numerical models' i.e. will this be examined in this paper based on the data and interpretations presented? If not, then this is a misleading statement and should be altered or removed. I do not see any point in making the observation that "It has been suggested that the northeastern part of the GIS reached the inner or middle parts of the continental shelf during its maximum extent during the last glacial (see Funder et al. 2011 for a review)" as more recent studies of Evans et al. 2009, O Cofaigh et al. 2004, Arndt 2018, Arndt et al. 2015, 2017, Arndt and Evans 2016, and Laberg et al. 2017 show quite clearly that ice went beyond the inner and middle shelf. The authors make this same point so there is no need to repeat an outdated debate. Include Evans et al. 2009, O Cofaigh et al. 2004, Arndt et al. 2015, and Arndt and Evans 2016 in the studies that have indicated ice was much more extensive on the NE Greenland shelf than the original summaries of Funder et al. 1998 and Funder et al. 2011 implied.
***We have altered the sentence regarding `validation and improvement of numerical models', emphasizing the need for paleo-reconstructions. Furthermore, we removed the "previously suggested maximum extent of the GIS in NE Greenland" by Funder et al. (2011), replacing it with more updated studies as suggested by the referee.***

The authors need to highlight how the swath bathymetric data presented in this paper differs to that presented in Laberg et al. 2017, and then detail how this study is different to that of Laberg et al. 2017. The same data for the outer shelf is presented again and there needs to be a clear statement or discussion differentiating what is published and what is new. I suggest that the authors add a section detailing what is known about the swath bathymetry and sub-bottom profiler data and implications for ice sheet history and sedimentary processes of the Laberg et al. 2017 study.

*We rewrote the introduction to chapter "4.2 Submarine landforms", providing information about what part of the data is new, and what has been previously published in Laberg et al. (2017). In the re-submitted version of the manuscript we clarify which part of the data set from Laberg et al. (2017) we have re-interpreted and why.*

Section 4.1 The range of analyses performed from geochemistry, sediment grain size, shear strength, etc. are outlined in the paper, but there is no reference to the actual data within the description of the lithofacies o, even in the interpretation of the lithofacies or the discussion. For instance, the 'magnetic susceptibility and Ca/Sum ratio vary between each core, with the highest in HH17-1326 and lowest in HH17-1328. Wet bulk density and shear strength are generally high: : :'. This is vague and does not serve the paper well. There is no subsequent use of much of this detailed data when it comes to the discussion of the glacial history later in the paper. I am still uncertain as to the point of including the magnetic susceptibility, XRF and wet bulk density data in this paper beyond including them for the sake of it. *We agree and have, therefore, taken out the XRF core scanner and shear strength data in the re-submitted manuscript.*

The interpretation of Facies 3 should explain what is meant by 'open conditions' and explain how the 'outer ice-proximal setting' inferred to be the location of the depositional environment differs from that envisaged for Facies 4. The paper notes the similarity of Facies 2 and 1 apart from the presence of IRD. Does this merely reflect the stochastic behaviour of icebergs rather than anything to do with permanent sea-ice or 'increased influence of drifting ice' in the sense of increased iceberg calving. The differences between the facies is essentially down to the vagaries of iceberg processes. *We rephrased these paragraphs and hope it is more clearly now (see page 5, lines 4-7, 15-19 and 28-32).*

Section 4.2 I am not convinced that there are MSGL in Figure 4 and 6. The features shown in Figure 5 appear to be lineations rather than MSGL and the description of them only refers to their length as >1.5 km. *We understand the reviewer`s point. However, we keep our suggestion that the landforms are fragments of/partly buried MSGLs, because the lengths/width ratios exceed 10:1 (cf. Clark, 1993).*

Do sub-bottom profiler records across the GZWs exist in order to rule out that they are bedrock sills? *We have included the publication by Petersen et al. (2015) showing that there is a thick Neogene sedimentary succession offshore NE Greenland, ruling out bedrock sills.*

Figures 4 and 5 are misleading as the recessional moraines and crevasse squeeze ridges are merged and have the same colour scheme, and it is difficult to distinguish where the crevasse-squeeze ridges are located. *Corrected.*

I am not convinced that some of the ridges represent a rhombohedral network indicative of crevasse-squeeze ridges. There appears to be little difference between the recession moraines and the crevasse-squeeze ridges apart from slight differences in morphology that might be linked to variations in grounding line processes and behaviour. The CSR appear to have a limited distribution and are not pervasive or widespread implying that the interpretation of 'surging' is unlikely and that they are more likely to be a localised feature maybe related to complex pattern of recessional moraines linked to ice-margin processes during standstill and retreat. Therefore, the idea of surging behaviour may not be correct and that the landform assemblages only record variable rates of grounding ice margin retreat and stabilisation. *We appreciate the extensive comment of the referee! Based on that, we revisited the data set and changed our interpretations from crevasse-squeeze ridges and multi-keel ploughmarks to saw-tooth moraines. Therefore, we rewrote the part of the result chapter regarding these specific landforms as well as the following discussion chapter.*

If indeed these features are CSR, why do they have to be associated with a surge rather than an advance/acceleration of an ice stream (linked to mass balance) and formation of basal crevasses due to tensile stress and ice break-up as it steps back to a stillstand position? Also, if it's a surge or even a simple readvance/acceleration of an ice stream, why aren't these features more widespread across the trough floor as presumably, a wider area would stagnate? The limited distribution implies a more complex recessional moraine pattern linked to complex ice retreat in some areas. ***See reply to comment regarding rhombohedral network and crevasse-squeeze ridges, above.***

I'm not convinced that the features identified as multi-keel iceberg ploughmarks is correct as they appear identical to the recessional moraines in Figure 4, 5 or 6. How would you even differentiate between a multi-keeled iceberg ploughmarks and the intervening ridges they create from those that are recessional moraines? ***See reply to comment regarding rhombohedral network and crevasse-squeeze ridges, above.***

Section 5 The authors state that "We propose that the Store Koldewey Trough was filled by grounded ice originating from the area presently covered with the Storstrømmen ice stream (Fig. 8A). This implies that the northeastern sector of the GIS reached a thickness allowing the ice stream to flow unrelated to the underlying topography, including the mountain ranges between present day Storstrømmen and Germania Land." This is speculative statement on its own. On what basis or geomorphological evidence are you making this assertion? Why wouldn't Storstrommen have preferentially flowed along and filled Dove Bugt Trough? The authors then go on to note that "An alternative interpretation is that Store Koldewey Trough had a much smaller drainage-basin, limited to Germania Land (Arndt et al., 2015). However, based on our data, including the observations of mega-scale glacial lineations, recessional moraines and grounding zone wedges, we favor the interpretation of Storstrømmen filling Store Koldewey Trough during full glacial conditions based on the volume of ice needed to fill a trough of this magnitude. We propose that the ice sheet thinned and that the underlying topography controlled the direction of ice flow during a late phase of the last glacial, i.e. that the ice flow from the interior of the GIS was directed to Jøkelbugten in the north and Dove Bugt in the south (Fig. 8B)." What is being proposed is speculative. Therefore, the discussion on the topographic and non-topographic controls on ice stream flow pathways, source and development from one to the other needs to be developed further. ***We have elaborated on this topic by providing information on the altitude that the Storstrømmen Ice Stream had to overcome to drain into Store Koldewey Trough (single peaks of 500-900 m), complimented with modelling results of paleo-ice sheet thickness on Germania Land during LGM (1000-1500 m; Fleming and Lambeck (2004) and Heinemann et al. (2014)).***

Why does the retreat of the grounded ice margin have to be 'slow' between stillstands? What evidence is used to support this assertion? There are no radiocarbon dates from the study cores that constrain ice stream retreat, so it is not possible to conclude the relative rate of retreat. Evidence from Antarctica shows that ice streams can abandon their groundling zone very quickly and then retreat at variable rates to the next stabilization point. It is worth exploring the issue of terrain factors (e.g. trough dimensions, trough depth distribution, underlying bed slope, etc.) modulating externally driven ice sheet retreat. The authors should consider the literature on GZW morphology and volume as an indicator of the relative length of time that the grounding line remains stable in one place (e.g. Dowdeswell et al. and Batchelor et al.). The authors need to develop the discussion in terms of what the smaller recessional moraines versus the larger GZWs mean for ice stream retreat rates, length of time of stabilisation and ice margin behaviour during temporary stillstands. For instance, the smaller moraines may be winter advances during stillstand.
***These are many good suggestions that we appreciate! The terms "slow" and "episodic" retreat have been introduced by both Ó Cofaigh et al. (2008) and Dowdeswell et al. (2008) discussing styles of ice retreat accompanied with the formation of recessional moraines and grounding zone wedges, respectively. We wish to continue using these terms and have, therefore, rephrased the paragraph, hopefully making our use of terms more clear to the reader. Furthermore, we provide a more in-***

***depth discussion of the relationship between local trough geometry and locations of the GZWs, as well as the sedimentary environments regarding the different lithofacies.***

The authors note that "We interpret the break-up and retreat of the GIS to have happened in two stages; initial retreat by breaking up and calving of grounded ice due to eustatic sea level rise caused by melting of ice at lower latitudes (Lambeck et al., 2014) (Fig. 9: Stage 2) and a second phase of melting driven by ocean warming, possibly due to the onset of inflow of intermediate water masses. The latter is supported by the occurrence of meltwater-channels and laminated sediments interpreted to be a result of excessive meltwater production in the middle and inner parts of the trough". On what basis, evidence or studies are you making this assertion for this region of Greenland, particularly the impact of sea level rise or inflow of intermediate water masses? What intermediate water masses are you referring to? There is no sediment evidence such as iceberg rafted lithofacies recorded in the cores to support iceberg calving and margin retreat due to sea level rise. Meltwater derived sediment facies cannot be used the defining piece of evidence indicating ocean warming retreat as the ice sheet will always produce and discharge meltwater due to the simple fact the ice at the subglacial bed is at pressure melt point. In fact, meltwater sediments will be deposited even when sea levels are rising and causing the ice margin to retreat.

The impact of these external factors will depend on the relative balance between atmospheric warming, precipitation, ocean warming and sea level rise, but stating that sea level rise causes a first stage of retreat is too simplified. For instance, studies in Antarctica show that maximum grounded ice extent in some sectors of the Ross Sea occurred during deglaciation even though there was atmospheric and oceanic warming and sea level rise because precipitation had a more dominate impact on mass balance, but eventually ocean factors dominated to cause retreat. How do you know there are two phases to the retreat of the ice sheet in this region without age constraints? In fact, the geomorphology implies more than two stages to ice sheet retreat. It is also worth noting that ice sheet retreat history is not merely a simple function of sea level rise, ocean warming and atmospheric warming but also due to terrain factors that can modulate ice sheet response and the rate of response to these external factors. The authors note the importance of the terrain for ice sheet retreat but do not really consider the literature that have looked at the impact of terrain factors on ice stream retreat. For example, Stewart et al. 2012 and Livingstone et al. 2012. These studies show the importance of trough width and depth on the rate of ice sheet retreat and try to quantify rates of retreat. ***Following the comments of the referee, we revisited the paragraphs mentioned above, and concluded that additional proxy information is needed to identify the driving forces causing the retreat of the GIS. In consequence, we have removed the paragraph.***

The authors note that "Based on the varying numbers of GZW$_0$s we suggest that retreat/ readvances of the ice streams offshore NE Greenland occurred asynchronously." Whilst I agree that it is possible that ice streams over such a large region as NE Greenland will experience asynchronous behaviour, I am not convinced of the evidence that is presented for this assertion. The data from Norske Trough, Westwind Trough and elsewhere do not provide a complete coverage of the respective areas and it is possible that there may be GZWs that exist, but have yet to be discovered undermining the suggestion that the number of GZWs indicates asynchronous ice stream behaviour. Secondly, without chronological constraints on regional ice stream behaviour during deglaciation or the ages of GZWs then the assertion of asynchronous behaviour is speculative. ***We agree that there might be undiscovered GZWs in Norske Trough and Westwind Trough. We rephrased this paragraph, focusing on the presentations of facts, rather than speculating on differences in deglaciation dynamics between different troughs.***

The authors note that "The present sub-glacial topography of Storstrømmen consists of a reversed bed slope, accompanied by a floating ice tongue (Hill et al., 2018). Thus, a potential future response to increased ocean warming could result in episodes of rapid retreat as the ice front undergoes thinning and/or ice tongue collapse. Such episodes are believed to cause a dynamic response up-glacier,

resulting in an accelerated ice flow, contributing directly to sea level rise (Hill et al., 2018)". It is not entirely clear how this statement links, and is relevant, to the previous paragraphs discussing ice sheet behaviour during deglaciation. ***We see the referee`s point and have removed this paragraph.***

The authors equate 'surge' behaviour with an ice stream. Why does the ice stream have to surge rather than simply readvance/accelerate? I am not convinced that the features they describe are crevasse-squeeze ridges but if they are then the section needs to be developed further to explain and justify why the ice stream surges as opposed to accelerate and readvance. The authors also need to explain why the CSR are limited in their spatial extent and distribution within the swath bathymetry dataset and why they have a close association with the GZW and recessional moraines. ***The landforms interpreted as surge-related landforms in the first version of the manuscript have been re-interpreted in the resubmitted version (see above), making the concept of surging irrelevant for the manuscript.***

---

## Author Comment (AC3) · 24 Jun 2020

Dear prof. Stokes!                                              Tromsø, June 24, 2020

First of all we would like to thank the reviewers for providing us a thorough and constructive review of our manuscript "Last Glacial ice-sheet dynamics offshore NE Greenland – a case study from Store Koldewey Trough". Their critical and constructive comments have been very helpful in improving the manuscript.

In the file *Revision Store Koldewey Trough – changes marked*, added and re-written paragraphs, as well as corrections of misspellings are marked with track changes.

In the following, we would like to give some feedback on the referees' comments and on some changes that we carried out as a consequence of the reviews.

Both reviewers commented on our use of the geophysical data previously published by Laberg et al. (2017), arguing we should be more explicit in differentiating what is published and what is new, as well as pointing out our re-interpretations. We agree that this should be stated more clearly and have, therefore, provided this information in chapter '4.2 Submarine landforms: glacial – deglacial ice-sheet dynamics' in the revised manuscript.

In the first version of the manuscript, we had included XRF core scanner data, shear strength, magnetic susceptibility and wet bulk density data in the result chapter. Both reviewers found our use of this data unsatisfying, suggesting we either remove it from the manuscript or discuss it further. We agree with the referees and have, therefore, removed the XRF core scanner and shear strength data from the manuscript, thus, focusing on the data that is absolutely relevant to determine the different lithofacies.

The reviewers questioned the landforms interpreted as crevasse-squeeze ridges and multi-keel iceberg ploughmarks in the first version of the manuscript. Therefore, we have re-visited the data set and examined existing literature once more. The result is that we changed the interpretations of these landforms to be saw-tooth moraines. As a consequence, the discussion regarding possible surging during the deglaciation has been removed. Both reviewers were unconvinced by our interpretation of the mega-scale glacial lineations on the middle shelf. However, we wish to keep our interpretation and have provided justifications for this in the comments to reviewers.

We have restructured and rewritten the discussion chapter. This includes reorganization of existing paragraphs, as well as incorporation of new paragraphs focusing on possible drivers of retreat, the role local trough topography may have had on the retreating ice front, as well as calculations on GZW volumes and the relative length of time of grounding line stabilization.

We removed figure 10, a schematic landform-assemblage model for Store Koldewey Trough, as suggested by the referees.

Both the abstract and conclusions have been adjusted according to the changes in the revised manuscript.

With the support of the co-authors of the manuscript, the author list has been rearranged, now listed by relative contribution.

We hope that the responses to the referees and the adjustments of the manuscript are satisfactory, and that the manuscript has a level to be accepted for publication in The Cryosphere.

Please to not hesitate to contact us in case of any further questions and/or the need of additional clarifications!

Kind regards on behalf of the authors,
Ingrid Leirvik Olsen

---

## Author Response (AR1)

[revised manuscript text omitted]
 the ice front has retreated and IRD becomes a more dominant component on the expense of suspension settling, compared to facies 4. (cf. Boulton 1990).

**4.1.4 Facies 2 – Massive glacimarine sediments (Fm)**

Massive olive gray to dark gray mud with little to moderate bioturbation and rare clasts composes facies 2. The facies is 10-80 cm thick and occurs in all four sediment cores (Fig. 2 and 3; Table 2). Both the lower and upper unit boundaries are gradational. The physical properties, including wet bulk density and,-magnetic susceptibility and shear strength-vary slightly within the facies and between the studied sediment cores.

The facies is interpreted to reflect suspension settling in an ice-distal glacimarine environment with limited iceberg or sea-ice rafting (Boulton and Deynoux, 1981). With increased distance from the grounding line, deposition from turbid meltwater plumes typically grade into more massive, bioturbated mud (Ó Cofaigh and Dowdeswell, 2001). However, itIt could also be speculated that there was a permanent sea ice cover during the deposition of this facies.

5 The rare amount of IRD within the massive mud may be a consequence of warm surface water during the Early Holocene causing prolonged open water conditions and reduced ice rafting on the shelf (Müller et al., 2012; Syring et al., 2020).

**4.1.5 Facies 1 – Massive glacimarine sediments with IRD (Fm (d))**

10 The uppermost facies in all of the studied cores consists of massive mud with clast-containing intervals-containing elasts, the latter interpreted to be IRD (Fig. 2 and 3; Table 2). The sSediment color alternates between brown to dark grayish brown, as well as olive gray to dark olive gray. Facies 1 is generally coarser than facies 2, with a pPeaks in the magnetic susceptibility-and decreases in Ti/Sum ratios\_corresponding to the depth with highest abundance of coarser material. The Ca/Sum ratios increase towards the top of the facies. Both tThe wet bulk density and shear strength areis similar to the underlying facies 2.

Facies 1 is interpreted to have been deposited in a similar environment as facies 2. Deposition of IRD can occur from dropping and dumping (see Vorren et al., 1983), i.e. dumping from a single iceberg or ice flow may be misinterpreted as enhanced iceberg rafting. However, since we identify increased amounts of IRD in all four cores we are confident that facies 1 reflects increased ice rafting at a regional scale, most probably related to the Neoglacial cooling trend (cf. Syring et al., 2020). However, the enhanced presence of clasts indicates an increased influence of drifting icebergs and/or sea ice.

**4.2 Submarine landforms: glacial – deglacial ice-sheet dynamics**

25 The swath bathymetry data from the middle and outer shelfpart of Store Koldewey Trough reveal glacigenic landforms interpreted to reflect various stages of ice-sheet extent, flow dynamics and retreat patterns. This is based on an entirely new data set from the middle shelf, in addition to including and expanding the data set described by (Laberg et al., (2017) from the outer shelf (Fig. 5). The addition of the new data set led to minor re-interpretations of landforms on the western part of the data set from (Laberg et al., (2017) (see subchapter 4.2.4. Curvilinear ridges – Saw-tooth recessional moraines).

**4.2.1 Streamlined landforms – mMega-scale glacial lineations (MSGL)**

Streamlined, trough-parallel grooves and ridges occur in the middle and outer trough-shelf (Fig. 4, 5 and 6A), terminating close to the shelf edge. Individual ridges have widths of 150-500 m and reliefs between 4-8 m (Table 3). They occur in clusters with spacing from 200 to 700 m. The grooves and ridges are partly eroded and/or overprinted by other landforms, which makes it difficult to measure their lengthsmaking the determination of their maximum lengths challenging. However, they appear to be Their minimum lengths range from >1.5 to >9km long with elongationand their length/width ratios of the ridges generally exceeding 10:1. The landforms occur in clusters with spacing from 200 to 700 m. High-resolution seismic data show reveal that the ridges are acoustically transparent (Fig. 7F), i.e. that their acoustic properties are a property that is characteristic for basal till (e.g. Ó

Cofaigh et al., 2007) <del>(Fig. 7F)</del>.

20

45

Based on the spatial distribution, dimensions and orientations, we interpret the grooves and ridges as mega-scale glacial lineations (MSGLs) formed subglacially at the base of a fast-flowing, grounded ice stream (Clark, 1993; King et al., 2009; Spagnolo et al., 2014) draining the GIS towards the shelf break. Similar landforms have been described on the seafloor of other formerly glaciated margins where they have been interpreted to indicate the

presence of grounded ice streams (e.g. Canals et al. 2000; Ottesen et al. 2005; Evans et al. 2009; Rydningen et al. 2013; Andreassen et al. 2014; Hogan et al. 2016; Arndt 2018).

**4.2.2 Large transverse ridges - Grounding zone wedges**

- 5 Four prominent bathymetric sills interpreted as grounding zone wedges A-D by Laberg et al. (2017) are present within the trough (Fig. 1B, C and 7). These authors presented acoustic data from wedges A and B, while the data from our study provides new information about wedge C. The wedges are 35-100 m high, 3.5-10 km wide and are spaced 45-60 km apart (Table 3). Sediment volumes per meter grounding line width are approximately 130 000 m3, 738 000 m3 and 150 000 m3 for wedges A, B and C, respectively. The cross-trough extent of the grounding
- 10 zone wedges exceeds the multibeam data coverage. Smaller ridges overprint the grounding zone wedges (Fig. 4, 5 and 6A). The base of the wedges is not possible to identify from our high-resolution Chirp profiles, but 2-D seismic profiles described by Petersen et al. (2015) reveal a thick Neogene sedimentary succession offshore NE Greenland, thus ruling out that these features are bedrock sills. As such, Tthese large landforms are interpreted to be produced by accumulations of sediments deposited at the ice stream's grounding line, recording the temporary
- 15 position of the grounding line during stillstand and/or readvance during a late phase of the last glacial in conformity with (Laberg et al., (2017).

**4.2.3 Small transverseridges – Transverse recessional moraines and crevasse-squeeze ridges**

The most prominent characteristic of the seafloor throughout Store Koldewey Trough is the high number of small ridges Multiple straight to slightly curvilinear transverse/semi-transverse ridges are visible on the seafloor of Store Koldewey Trough (Fig. 4, 5, 6 and 7). These ridges are one order of magnitude smaller than the grounding zone wedges and occur in two forms; (1) as curvilinear to straight transverse/semi transverse ridges, or (2) as rhombohedral ridge patterns. The ridges are up to 2200 m wide, have reliefs <50 m and have a spacing of 50-500 m (Table 3). Some of the ridges superimpose others, implying several generations of ridge formations. There are two generations of ridges on the outer shelf (between grounding zone wedges *A* and *B*), where the first generation is spaced ~80 m apart, whilst the superimposing ridges are larger and mostly spaced 200-400 m apart. The spacing of ridges on the middle shelf is commonly 100-200 m. The transverse ridges located southeast seaward of grounding zone wedge *C* appear to be spaced further apart and/or are less well preserved.

- 30 We interpret the curvilinear to straight ridges as recessional moraines formed at the grounding line during slow overall retreat with repeated stillstand and/or small readvances (cf. Dowdeswell et al., 2008; Ó Cofaigh et al., 2008). The rhombohedral network of ridges are interpreted to be crevasse squeeze ridges that formed from soft sediments squeezed into basal crevasses of the ice stream during and after its transition from fast flow to stagnation, often associated with glacial surges (Boulton et al., 1996; Dowdeswell and Ottesen, 2016; Evans and Rea, 1999;
- 35 Sharp, 1985; Solheim, 1991). The ridge-like features identified on the sub-bottom profiles are acoustically transparent suggesting a diamictic composition (Stewart and Stoker, 1990) (Fig. 7).

4.2.4 Multi-keel iceberg ploughmarks 4.2.4 Curvilinear ridges - Saw-tooth recessional moraines
 Linear to cClusters of curvilinear depressions with berm ridges along their lateral marginsridges occur in clusters,
 characterizinge the seafloor east both landwardproximal and seaward of grounding zone wedge *B*, and seaward of
 grounding zone wedge *C* (Fig. 4, 5, 6B, E- and 7D, E). The depressions are flat bottomed and occur often in
 groupsIn planview the curvilinear. These ridges often-occur often closely spaced, exhibiting a saw-tooth pattern in
 plan view. Many of the features continue as long moraine ridges oriented sub-parallel withto the ice-flow direction.
 Bifurcations and cross-cutting patterns occur. Individual depressions with associated-ridges are up to 1.3 km long,
 170–1100 m mide and have a mide of 5, 20 m (Table 2).
 The deministration of the deministration of the second moraine flat bottomed and provide and patterns occur.

45 170-1100 m wide and have a relief of 5-30 m (Table 3). They are predominantly orientated parallel to the trough axis. They are typically asymmetrical with a steeper ice distal slope and a more gentle ice proximal slope (Fig. 6E). The depressions saw-tooth ridges partly superpose and thereby obscure reworkedmodify the underlying

transverse ridges and creatinge a chaotic seafloor patterns.- Furthermore, grounding zone wedge *B* partly covers some saw-tooth ridges.

We interpret the saw-tooth ridges as recessional- moraines that formed by a combination of push- and squeeze processes, recording an active ice retreat punctuated by periodic advances-. The formation of these distinctive landforms is inferred to be dependent on the topography, where down-ice widening, in this case of the trough, causes increased transverse stress leading to longitudinal crevasses initiating an irregular ice front-). Similar saw-tooth like moraines have been observed in e.g. Norway (Burki et al., 2009; Matthews et al., 1979), Barents Sea (Hogan et al., 2010; Kurjanski et al., 2019), Iceland (Chandler et al., 2016; Evans et al., 2016) and Arctic Canada

10 (Andrews and Smithson, 1966). The landforms were previously interpreted as rhombohedral ridges by (Laberg et al., (2017) on the western part of the data set from the outer shelf, however, we found the saw-tooth-like morphology incompatible with the geometric ridge networks of rhombohedral ridges (cf. Bennett et al., 1996).

We interpret these landforms as plough marks generated from grounded icebergs with multiple keels. The groups
 with parallell ploughmarks are comparable to the features suggested to be a result of multi-keeled icebergs in e.g. West Antarctica (Wise et al., 2017) and northern Barents Sea (Andreassen et al., 2014). However, the ploughmarks in this study are one magnitude shorter and corrugation ridges within the furrows appear to be absent. We suggest both the uniform orientation and limited length of the ploughmarks to be a result of the presence of an ice melange limiting iceberg drift, similar to the observations of similar landforms by Kristoffersen et al. (2004) in the Arctic Ocean.

**4.2.5 4.2.5 Straight incisions - Channels**

Two straight incisions that are U-shaped in cross section, 150-300 m wide and with incision depths of 3-10 m are identified along the northern and southern trough sidewalls (Fig. 4; Table 3). The incisions are oriented parallel to the recessional moraines and continue beyond the extent of the swath bathymetry data set. They cut into the mega-scale glacial lineations and the acoustically transparent sediments interpreted as basal till (see below for description and interpretation of the latter). The landforms are interpreted as channels formed during deglaciation and are probably related to erosion by meltwater, as the ice sheet disintegrates and produces fractures filled with meltwater eroding beneath the ice sheet. Another possible explanation for their formation is they could have formed from meltwater runoff from ice masses remaining on the surrounding banks.

**4.3 Seismostratigraphy**

Two seismostratigraphicie units (S1 and S2) were identified in the chirp sub-bottom profiles in Store Koldewey Trough (Fig. 7).

35

**4.3.1 Unit S1 – Glacigenic deposits and/or sedimentary bedrock**

Seismic unit S1 is the lowermost seismostratigraphic unit and the base of this unit represents the acoustic basement occuring inof the entire study area. IThe unit has an acoustically transparent to semi-transparent signature and an irregular top reflection with medium to high amplitude and continuity (Fig. 7B).

40

The unit correlates with lithological unit 5 (*Dmm*) in the sediment cores, interpreted as subglacial till, i.e. that it includes subglacial deposits. However, the chirp profiles (Fig. 7) reveal that the unit S1 also includes grounding-zone wedges, as well as transverse and rhombohedral ridgesrecessional moraines, i.e. multiple glacigenic landforms and deposits. In the majority of the study area, these glacigenic landforms define the acoustic basement.

45 However, a relatively strong and smooth reflection can be observed beneath glacigenic deposits. This is interpreted to be caused by the top of the underlying bedrock, suggesting that S1 also includes bedrock in some areas.

**4.3.2 Unit S2 – Latest Weichselian - Holocene gGlacimarine sediments**

Unit S2 is an acoustically transparent unit (Fig. 7). The unit is thin and only occurs locally either as an infill between the topographic highs or draping the underlying unit S1, i.e. it isor missing from most of Store Koldewey Trough (e.g. on the outer shelf), with a The maximum thickness of the unit is 2.5 m where present (Fig. 7). The

5 sediment unitIt occurs locally either as an infill between the topographic highs or draping the underlying unit S1.

Unit S2 is sampled with all four sediment cores and is interpreted to include correlated with the lithological units 4 (*Fl*), 3 (*Fl*(*d*)), 2 (*Fm*) and 1 (*Fm*(*d*)), i.e. it occurs at all four core sites. The unit is identified ascontains glacimarine deposits reflecting, consisting of a gradual transition from glacier proximal to distal glacimarine sediments from the latest Weichselian at the base to distal glacimarine Holocene sediments at the top-environments.

**5 Discussion**

10

15

20

**5.1 Maximum ice sheet extent and influence of subglacial topography**

[revised manuscript text omitted]

trough shallows towards the coast. A combination of There are several factors that possibly can may have preconditioned this, and led to the complex geomorphology in Store Koldewey Trough: i) local highs in an the overall shallowing landward seafloor profile may have provided pinning points, causing ice stabilization and promoting longer stillstands during the deglaciation; ii) trhough narrowing towards the coast may have

- 5 lateral stress on the retreating ice margin, thus slowing down/stabilizing ice flowt; iii) repeated advances due to glacial surges during deglaciation based on the documented grounding zone wedges A and B accompanied by erevasse fill ridges; or iv) the GIS in Store Koldewey Trough possibly had a more dynamic response to the changing climatic and oceanographic conditions compared to troughs of similar dimensions elsewhere on the NE Greenland Margin. The grounding zone wedges in Store Koldewey Trough occur in areas of relatively marked
- 10 decreased water depths and reductions in trough width (Fig. 1), indicating that local through geometry led to a repeated re-stabilization of the grounding line. Because the formation of grounding zone wedges and recessional moraines require a grounded ice stream/glacier margin we exclude a rapid/continuous retreat by ice stream flotation.
- 15 Whereas many paleo-ice streams on other glaciated continental shelves with reverse bed slopes experienced a lift-off-from the seafloor and an initial rapid-retreat due to sea-level rise, e.g. Norske Trough (Arndt et al., 2017), Amundsen Sea in West Antarctica (Smith et al., 2011) and NW Fennoscandian ice sheet (Rydningen et al., 2013)), the ice stream in Store Koldewey Trough stayed grounded or repeatedly stabilized as the trough shallows towards the coast. Consequently, Store Koldewey Trough has a more complex landform assemblage than other ice stream settings on formerly glaciated continental shelves (Fig. 10).

The locations and dimensions of grounding zone wedges on the NE Greenland continental shelf have up to recent years been are thus far poorly documented. Batchelor and Dowdeswell (2015), and referring to Dowdeswell and Fugelli (2012), mention six grounding zone wedges in Store Koldewey Trough based on IBCAOon seismic data.

- 25 However, our wOur data set reveals provides evidence for the existence of only four GZWsgrounding zone wedges. Large-Additional grounding zone wedges have been documented from other cross-shelf troughs in the region (Arndt, 2018; Arndt et al., 2015, 2017; Evans et al., 2002; Winkelmann et al., 2010). These studies reveal, however, the occurrence of only single grounding zone wedges, e.g. in Norske Trough and Westwind Trough (Arndt et al., 2015; Winkelmann et al., 2010). Arndt et al. (2017) suggest that the grounding zone wedges in Norske
- 30 Trough and Westwind Trough formed as the GIS readvanced during the Younger Dryas. Based on the varying numbers of GZW's we suggest that retreat/readvances of the ice streams offshore NE Greenland occurred asynchronously.
- The formation of grounding zone wedges typically requires a stabilization of the ice margin for decades to centuries (Dowdeswell and Fugelli, 2012) (Fig. 9: Stage 4). This period can be estimated when sediment flux across the grounding line and grounding zone wedge volume are known (Howat and Domack, 2003). Grounding zone wedges *A* to *C* in Store Koldewey have volumes of approximately 130 000 m3, 738 000 m3 and 150 000 m3 per meter grounding line width. In the absence of chronology we apply sediment flux of 102 to 103 m3 m-1 yr-1 to the grounding line, as calculated for other paleo ice streams in Greenland (Hogan et al., 2012, 2020), the West
- 40 Antarctic Ice Stream (Anandakrishnan et al., 2007) and on the southern Norwegian continental margin (Nygård et al., 2007). Applying these numbers suggests that the formations of grounding zone wedges *A*, *B* and *C* took at least 130, 740 and 150 years, respectively.
- The recessional moraines are generally one to three orders of magnitude smaller than the grounding zone wedges.
   Accumulations of retreat moraines have repeatedly been referred to as 'annual moraines' correlated with annual cycles including winter advances and summer retreats during the overall deglaciation (Baeten et al., 2010; Boulton, 1986; Kempf et al., 2013; Ottesen and Dowdeswell, 2006) (Fig. 9: Stage 3). Assuming that accumulations of retreat moraines reflect annual moraines, we propose the following deglaciation velocities in the study area: following the formation of grounding zone wedge A at the shelf edge, the grounding line retreated with an average of 80 m yr-1 before accelerating to 200-400 m yr-1 on the outer shelf (Fig. 9: Stage 6). By the time the ice margin

The present sub-glacial topography of Storstrømmen consists of a reversed bed slope, accompanied by a floating ice tongue (Hill et al., 2018). Thus, a potential future response to increased ocean warming could result in episodes of rapid retreat as the ice front undergoes thinning and/or ice tongue collapse. Such episodes are believed to cause a dynamic response up glacier, resulting in an accelerated ice flow, contributing directly to sea level rise (Hill et al., 2018).

5 <del>al., 201</del>

The lithological sequence starting with a basal till overlain by glacimarine deposits suggests the transition from sub-glacial to ice-proximal and, subsequently, to a more ice-distal environment dominated by suspension settling with various degrees of ice rafting. Evans et al. (2002), and Smith et al. (2011) and (Reilly et al., (2019) documented sedimentological facies with similar characteristics from the deglaciation of the trough offshore Kejser Franz Josef Fjord, and the West Antarctica Ice Sheet and Petermann Glacier, respectively, implying that they recorded the transition from a grounded ice sheet to open marine environments. The deglacial lithofacies (3 and 4) reflect different depositional environments (Table 2): whereas the influence from meltwater was stronger during the deposition of facies 4, the supply of IRD was higher during the deposition of facies 3. The lack of IRD in an ice proximal setting may have several explanations; i) the time of deposition may represent a period with an extensive sea-ice cover preventing icebergs to drift over the area (Jennings and Weiner, 1996; Moon et al., 2015; Warren and Placeer 2002). iii) a high fluen of an along a place p

- Vorren and Plassen, 2002), ii) a high flux of sediment-laden glacial meltwater mask the amount of IRD (Boulton, 1990) or iii) the sediments may be deposited in a sub-shelf environment far enough from the grounding line to be unaffected by mass flows and rain-out of basal debris (Domack and Harris, 1998; Jennings et al., 2019; Smith et
- 20 al., 2017). Absence of basal debris in the ice stream seems unlikely given the underlying basal till (facies 5) and overlying facies 3 abundant of clasts and IRD. Nevertheless, Reilly et al. (2019) provided evidence of an IRD free depositional environment beneath the former ice tongue of Petermann Glacier in NW Greenland, with a following increase in IRD concentrations as the ice tongue retreated from the site. We note that facies 4, characterized by lamination and the absence of clasts, occurs exclusively in the two cores on the inner shelf. Given that the coring
- 25 sites are located within depressions, it could be assumed that the ice detached from the ground leading to sub-ice shelf environments where deposition was dominated by suspension settling (compare with Reilly et al., 2019). We speculate that the trough narrowing towards the coast contributed to an increase in lateral drag and subsequent reduction in extensional stress as the ice front retreated to the inner shelf, resulting in a more stabilized ice front and ice-shelf formation (Fig. 9: Stage 5). The deglacial lithofacies (3 and 4) reflect different depositional
- 30 environments (Table 2): whereas the influence from meltwater was stronger during the deposition of facies 4, the supply of IRD was higher during the deposition of facies 3. The presence of facies 4 exclusively in the two westernmost cores suggest that either the style of retreat was different between the middle and inner part of the trough, or the deglacial lithofacies deposited during the initial ice retreat was removed from the middle trough area through winnowing. However, the identification of subglacial channels in the middle part of the trough indicate that meltwater was present.

**5.3 Postglacial development**

During the late phase of the deglaciation After the deglaciation of Store Koldewey Trough, the ice stream retreated across Store Koldewey Island and Germania Land (Fig. 9: Stage 96)., terminating This terminated the supply of suspended sediment and icebergs to Store Koldewey Trough, and delivering icebergs and meltwater re-routing the material to Dove Bugt and Jøkelbugten (Fig. 8B) instead. This resulted in the termination of sediment input from this sector of the GIS to Store Koldewey Trough, which The change of ice configuration and sediment supply may explain the thin sediment drape on top of the glacigenic deposits (Fig. 7).

45

Postglacial sedimentary processes in the trough are interpreted to comprise hemipelagic deposition of terrigenousestrial material fromin sea ice transported southwards from the Arctic Ocean with the Transpolar Drift, rainout from icebergs and meltwater plumes released from regional marine terminating outlet glaciers north of the study area (e.g. 79°-Glacier and Zachariae Isstrøm), in addition toas well as winnowing on the surrounding hanks and resumption of the finant reation within the unpermost lithological unit.

50 banks.and resuspension of the finest sediment fraction within the uppermost lithological unit.

The low IRD content in facies 2 is probably due to reflects a combination of multiple factors: i) ice fronts retreating from the marine realmon land, ii) while-material entrapped in icebergs from 
[revised manuscript text omitted]

- 10 Buizert, C., Gkinis, V., Severinghaus, J. P., He, F., Lecavalier, B. S., Kindler, P., Leuenberger, M., Carlson, A. E., Vinther, B., Masson-Delmotte, V., White, J. W. C., Liu, Z., Otto-Bliesner, B. and Brook, E. J.: Greenland temperature response to climate forcing during the last deglaciation, Science, 345(6201), 1177–1180, doi:10.1126/science.1254961, 2014.

[revised manuscript text omitted]

---

## Referee Report (RR1)

**Olsen et al.: Last Glacial ice-sheet dynamics offshore NE Greenland – a case study from Store Koldewey Trough**

This manuscript is a considerably improved re-write/re-submission of a previously reviewed manuscript. The data are more clearly presented, the geomorphological interpretations are more logical (both with respect to morphological indicators and interpretations of glaciodynamic contexts/conditions for particular landform types), and the Discussion is better formulated. That said, I still have some concerns about the rigour of data interpretation and justification for some of the elements of the Discussion.

*Sediment cores*

The presentation of core data seems limited. The text reporting is clear and logical but I wonder if more use of valuable core material could have been developed. Was any grain size sorting analysis done? – you have quantitative data on grain size fractions. Why are clast proportions/abundance not quantitatively reported, since you've sieved clasts out as a specific size and so presumably have these data? What are the colour changes related to – organic content? Is 'high' or 'low' density just on relative terms (to the rest of the core/s) or compared to literature typical values? – does 'high' bulk density necessarily mean over-consolidation of tills?

Facies 3 & 4 lack any references to other literature that would guide or support your interpretations (and the interpretations of the other facies are also sparsely referenced). Yet there is a wealth of work on proximal to distal glaciomarine sedimentary characteristics and physical properties. I suggest (all of) your interpretations should be grounded in available literature.

Could facies 3 mark the calving front of an ice shelf (melt out of englacial debris), as often suggested for first emergence of IRD after limited sub-ice shelf sedimentation in Antarctic deglacial/glaciomarine facies succession models? Are there other indicators that IRD would be from freely floating icebergs on an at least partly open ocean?

*Geomorphology*

The lineations are sparse, and I still question the 'mega-scale' interpretation; interpretation of an ice stream footprint shouldn't rest on one ratio, but rather a whole suite of landform and landform assemblage observations. Flutes on a valley glacier forefield can have elongation ratios of >10:1 – it doesn't make them msgls. At best, the evidence for fast flow here is limited to the outer shelf, and the discussion of this system should reflect the distribution of these limited indicators.

GZW A looks no different, morphologically, to the first two larger moraines inland of wedge A (about 15 km and then a further 5 km) – based on their appearance in Figure 5. Why interpret it as a wedge?

I would think there's some value in discussing overall trough morphology (orientation, depth, width, tributaries compared to other troughs along this margin), either as part of the geomorphological results or within the Discussion (part 5.1). This trough seems, from Fig 1, to have an abrupt start and a lack of obvious feeder tributaries – how does this relate to the discussion about the source of ice to the trough, or the flow velocity, sediment flux, erosional vigour?

*Discussion*

The Discussion is better focussed and internally logical, but some passages still are rather underdeveloped. Section 5.2 (dynamics during deglaciation), in particular, comprises a set of paragraphs that clearly fit within this theme but don't really connect to or build on each other, they just appear as discrete ideas. Try to better weave a discussion together from paragraph to paragraph.

The attempt to quantify grounding line landform formation times and retreat rates is a useful addition to the work, though the authors MUST acknowledge the assumptions and caveats implicit in the approaches that they take, and give ranges for their estimates that reflect the uncertainties in the approach; it is difficult to achieve much better than order of magnitude results. The Abstract and Conclusions are too definitive and the Discussion neglects important caveats. The use of any sediment flux makes considerable assumptions about sediment supply, mode of transport, ease of transport, thermal regimes, ice flow velocity. Are the sources you take your upper and lower fluxes from describing glacial systems that would be appropriate analogues for Storstrømmen? How well do these earlier studies really know (measure? infer? with what independent assumptions?) what their sediment fluxes are? Similarly, the interpretation of some types/arrangements of recessional moraines as annually forming is a major and controversial assumption. Annual formation isn't even straightforward to conclude (debate still continues) in classical regions where there are annually-resolved varve chronologies to inform the retreat pattern and rate. It is inappropriate to simply take this assumption without at least acknowledging the debate – and, preferably, discussing the validity of the choices you make.

I don't think that blindly applying conceptual interpretations of retreat *rate* like 'slow', 'rapid' or 'episodic' based on data that is inherently pattern-based rather than chronological, while acknowledging that you lack any chronological constraint, is especially constructive. Wishing to 'continue using the terms' put forward from previous conceptual work isn't a valid reason to do so – are they appropriate, given your data? Can you use your own evidence to evaluate whether these conceptual models are applicable, rather than just 'choosing' to apply them? I think you have more to say from your own data than just adopting a term that doesn't adequately summarise the range of retreat behaviours you see. Regardless of the interpreted retreat *rate* terminology, I would suggest it's more interesting that in a trough system, commonly expected to host well-formed lineations and retreat from wedge to wedge, here you have a record dominated by retreat landforms of various morphologies, marking different grounding line sedimentation processes and/or rates, different grounding line durations, and that overall the magnitude of the retreat events is rather small. The atypical landform assemblage for a trough setting is more interesting to explore than applying a conceptual model of rate that you can't say much about.

**Line edits**

Title: ice-sheet is hyphenated here, but nowhere else in the text.

*Abstract*
p1 line 10-12: re-consider the phrasing here in light of uncertainties/assumptions in calculations (see main comment in Discussion text).

p1 line 13-14: what evidence do you have that ice retreated directly across Store Koldewey Island and Germania Land? You can argue that at the ice sheet's maximum extent, ice was sufficiently thick to flow across this topography, but in a late stage of deglaciation? Isn't it more plausible that the

high ground forced flow paths (and retreat) around the topography? The latter is in fact what you conclude. This part of the abstract should be revised.

*Main text*

p1 line 20: "…GIS is presently drained…"

p2 line 14: explain why an absolute chronology is relevant to the previous sentence: chronology enables us to understand rates of change, while absolute ages let us tie retreat events in with external forcing (climate or ocean changes)

p2 line 17-19: do these cosmogenic dates record ice retreat, as in movement of the ice *front* landward of the island, or do they record ice sheet *thinning?*

p3 line 1: it deepens seaward but with bumps and dips along its length – it's not a smooth/steady deepening.

p3 line 5-8: while interesting, it doesn't seem relevant to the rest of the paper that this ice stream has displayed two recent surges.

p3 line 20: suggest ending this section with some sort of motivation statement for the rest of your work, or introductory statement to what you've done that builds on these previous studies that you mention.

p3 line 33: suggest here you comment on which data have already been published in Laberg et al, and which are new (i.e. move from p5 line 36-38). There also is a bit of a methods gap between this paragraph and the next, where you explain GZW volume estimates, despite not having mentioned that you have / how you have recorded GZWs. I think a sentence or so stating that you've mapped landform outlines or crestlines is needed, and perhaps the basis for how you've interpreted the environmental origins and how/why you've re-interpreted some earlier published ideas (e.g. assessment of size, shape, arrangement, sedimentary setting…)

p3 line 35: what do you mean by 'box volumes'? You've assumed a rectangular cross-profile? On what basis? Is this more valid than an asymmetric triangle?

p3 line 38: you could refer to other works that have followed a similar approach…

p4 line 2: you could note that this is a common problem in cold polar waters close to ice sheet grounding lines (relatively little life at the grounding line and low carbonate preservation subsequently (dissolution)) – plenty of other Greenland and Antarctica studies suffer the same limitations

p5 line 6: 'more dominant component AT the expense of…'

p5 line 24: refer to your sand-silt-clay data, specifically, to support that facies 1 is coarser than facies 2… although, looking at these plots, the coarser nature of facies 1 is not particularly convincing.

p5 line 26: suggest you note in the facies 1 and facies 2 observations that these two facies alternate in two of the cores – there may be more than one occurrence of each/either facies in the core succession.

p5 line 35: both Fig 4 and Fig 5 should be referenced here, not only Fig 5. E.g. '…data from the middle (Fig 4) and outer (Fig 5) shelf of…'

p6 line 14: what data did Laberg et al interpret wedges A-D from? Presumably not all multibeam, since you present a new block here? Seismic?

p6 line 17-18: can you give more information on how you calculated these volumes? Did you choose just one cross-profile per wedge, or several, and on what basis? From the multibeam, wedge C looks considerably larger than A, more comparable to B.

p6 line 31-32: a figure and more detailed presentation and discussion is warranted here if there are genuinely two superimposed sets of moraines. I can't see that this is evident from either the multibeam panel or the mapped interpretation of moraines in Fig 5. On what basis do you interpret superimposition? If there is, in fact, one group that sits clearly on top of another, then we must interpret that there's been a readvance: you would have one retreat assemblage buried under another. And in that case, this finding would warrant further discussion.

p7 line 10: describe or explain why you find the sawtooth-like pattern incompatible with your earlier interpretation.

p7 line 14: '…identified along the … sidewalls' sounds like channels are running parallel to the walls of the trough. These simply occur at the periphery of your data coverage, and cut obliquely through other landforms. Suggest you rephrase.

p7 line 37: either break the sentence after the list of lithological units ('… and 1 (Fm(d)). It occurs at all…') or insert 'and' before 'it'.

p8 line 10-13: suggest you switch these two sentences around, it flows more logically from the previous paragraph to begin with the till, and then what rests on top of the till.

p8 line 25: 1000-1500m thick, or high? (i.e. surface altitude or ice thickness?)

p8 line 30: since Storstrømmen is an outlet of the contemporary NEGIS, it sounds strange to talk about this as a 'similar flow feature'. Can you instead emphasise here that the disregard for topography appears (from your results) to be a characteristic of both the palaeo and contemporary NEGIS? And be specific about exactly where this independence from topographic steering occurs within the NEGIS today.

p8 line 31: 'a palaeo-ice stream' – can you be specific, which one?

p8 line 34-38: do Arndt et al propose a more restricted ice margin as well as a limited drainage basin (supply), or do they also suggest shelf-edge glaciation? If they envisage shelf-edge glaciation, then your counter-argument ('we have evidence of shelf-edge glaciation') isn't really sufficient.

p8 line 37: can you be more specific about the volume of ice required, and compare to what Germania Land could sensibly supply?

p8 line 42-45: I don't think these opening sentences really add anything useful.

p9 line 5: the phrasing here makes it sound like Store Koldewey also has a reverse slope. The fact that it doesn't is surely an important reason for any contrast in behaviour. You set up a 'problem' here that isn't really one. I would revise this paragraph as a commentary on the rather uncommon situation of having a seaward dipping trough that has led to a rather stable retreat pattern, supplemented by local trough shallow/narrow points, rather than making this more of a puzzle than it really is.

p9 line 13-14: what do you mean by 'had a more dynamic response to…'? Rephrase to say something direct, this is vague.

p9 line 16: 'local trough geometry' (typo, through)

p9 line 22: where are the other 2 wedges that Batchelor & Dowdeswell find here, and why do you not include those in your reconstructions?

p9 line 20-27: this paragraph doesn't seem to go anywhere. What do you interpret to be the significance in the number of wedges recorded in different troughs? Are the single wedges in Norske & Westwind Troughs at the shelf break? What would be the implications for your work if these formed during the Younger Dryas?

p9 line 35: which West Antarctic ice stream? Or do you mean ice sheet?

p9 line 37: these values use the upper sediment flux rate. Using the lower flux would give you an order of magnitude longer formation time, i.e. 1300, 7400 and 1500 years. Is there enough time available for retreat across this shelf, with those standstill durations? See also main comment about sediment flux assumptions.

p9 line 40: this passage must better reflect the debate about whether recessional moraines can be interpreted as annual or not.

p9 line 50-52: this sentence either could be removed (since you don't put it in the context of your results) or should reflect the vast literature on grounded to open marine (deglacial) sedimentological facies. Picking a random three papers that have studied this succession is rather meaningless.

p9 line 6: masks

p9 line 7-9: Prothro et al (Marine Geology) discuss the distance for rainout of basal debris distal to the grounding line – they find it to be extremely short.

p9 line 25: it is intriguing that the sediment drape only occurs across the inner-middle shelf, and not the outer parts that deglaciated first. Why do you think this is? Is this a supply or a preservation/deposition question?

p9 line 30: what is the significance of facies 1 & 2 alternating in the two outer cores? How does this affect your environmental interpretations here?

p9 line 37: would reduced sea ice not allow more icebergs to access the area? Or, conversely, expanded sea ice would limit access of icebergs and deposition of IRD?

Figure 1: label the profile shown in panel B on the dashed white line in panel A (instead of or as well as writing in the caption).

Figure 5: could you make the colours for sawtooth moraines and iceberg ploughmarks more distinct from one another?

---

## Author Response (AR2)

*The authors' replies on the comments are written in red and italics.*

Olsen et al.: Last Glacial ice-sheet dynamics offshore NE Greenland – a case study from Store Koldewey Trough

This manuscript is a considerably improved re-write/re-submission of a previously reviewed manuscript. The data are more clearly presented, the geomorphological interpretations are more logical (both with respect to morphological indicators and interpretations of glaciodynamic contexts/conditions for particular landform types), and the Discussion is better formulated. That said, I still have some concerns about the rigour of data interpretation and justification for some of the elements of the Discussion.

*First of all we would like to thank the referee for a thorough and constructive review, contributing to the improvement of the manuscript. It has been very helpful and we are grateful! We have responded on all comments by the referee below, as well as adding a list of relevant changes made in the revised manuscript.*

Sediment cores

The presentation of core data seems limited. The text reporting is clear and logical but I wonder if more use of valuable core material could have been developed. Was any grain size sorting analysis done? – you have quantitative data on grain size fractions. Why are clast proportions/abundance not quantitatively reported, since you've sieved clasts out as a specific size and so presumably have these data? What are the colour changes related to – organic content? Is 'high' or 'low' density just on relative terms (to the rest of the core/s) or compared to literature typical values? – does 'high' bulk density necessarily mean over-consolidation of tills? *We believe that we have provided sufficient core data for distinguishing sediment genesis and that taking the core data analysis one or several steps further would be outside the scope of this paper. We hope that this is acceptable.*
*Regarding the bulk density – we have specified that "high" density means relative to the rest of the core.*

Facies 3 & 4 lack any references to other literature that would guide or support your interpretations (and the interpretations of the other facies are also sparsely referenced). Yet there is a wealth of work on proximal to distal glaciomarine sedimentary characteristics and physical properties. I suggest (all of) your interpretations should be grounded in available literature. *Additional references have been included.*

Could facies 3 mark the calving front of an ice shelf (melt out of englacial debris), as often suggested for first emergence of IRD after limited sub-ice shelf sedimentation in Antarctic deglacial/glaciomarine facies succession models? Are there other indicators that IRD would be from freely floating icebergs on an at least partly open ocean? *We appreciate the suggestion and have included it to our interpretations.*

Geomorphology

The lineations are sparse, and I still question the 'mega-scale' interpretation; interpretation of an ice stream footprint shouldn't rest on one ratio, but rather a whole suite of landform and landform assemblage observations. Flutes on a valley glacier forefield can have elongation ratios of >10:1 – it

doesn't make them msgls. At best, the evidence for fast flow here is limited to the outer shelf, and the discussion of this system should reflect the distribution of these limited indicators. *We see the reviewers point and have opened for the lineations on the middle shelf to be interpreted as ´glacial lineations´.*

GZW A looks no different, morphologically, to the first two larger moraines inland of wedge A (about 15 km and then a further 5 km) – based on their appearance in Figure 5. Why interpret it as a wedge? *GZW A has a more wedge-shaped form when looking at the seismic data, as well as being magnitudes larger than the moraines further inland.*

I would think there's some value in discussing overall trough morphology (orientation, depth, width, tributaries compared to other troughs along this margin), either as part of the geomorphological results or within the Discussion (part 5.1). This trough seems, from Fig 1, to have an abrupt start and a lack of obvious feeder tributaries – how does this relate to the discussion about the source of ice to the trough, or the flow velocity, sediment flux, erosional vigour? *We have rewritten parts of the discussion chapter and included what we believe is appropriate given our data, without being too speculative.*

Discussion

The Discussion is better focused and internally logical, but some passages still are rather underdeveloped. Section 5.2 (dynamics during deglaciation), in particular, comprises a set of paragraphs that clearly fit within this theme but don't really connect to or build on each other, they just appear as discrete ideas. Try to better weave a discussion together from paragraph to paragraph.

The attempt to quantify grounding line landform formation times and retreat rates is a useful addition to the work, though the authors MUST acknowledge the assumptions and caveats implicit in the approaches that they take, and give ranges for their estimates that reflect the uncertainties in the approach; it is difficult to achieve much better than order of magnitude results. The Abstract and Conclusions are too definitive and the Discussion neglects important caveats. The use of any sediment flux makes considerable assumptions about sediment supply, mode of transport, ease of transport, thermal regimes, ice flow velocity. Are the sources you take your upper and lower fluxes from describing glacial systems that would be appropriate analogues for Storstrømmen? How well do these earlier studies really know (measure? infer? with what independent assumptions?) what their sediment fluxes are? Similarly, the interpretation of some types/arrangements of recessional moraines as annually forming is a major and controversial assumption. Annual formation isn't even straightforward to conclude (debate still continues) in classical regions where there are annually-resolved varve chronologies to inform the retreat pattern and rate. It is inappropriate to simply take this assumption without at least acknowledging the debate – and, preferably, discussing the validity of the choices you make. *We appreciate the extensive comment of the reviewer! Based on that, the revised manuscript is now less definitive in its conclusions, as well as acknowledging uncertainties.*

I don't think that blindly applying conceptual interpretations of retreat rate like 'slow', 'rapid' or 'episodic' based on data that is inherently pattern-based rather than chronological, while acknowledging that you lack any chronological constraint, is especially constructive. Wishing to 'continue using the terms' put forward from previous conceptual work isn't a valid reason to do so – are they appropriate, given your data? Can you use your own evidence to evaluate whether these conceptual models are applicable, rather than just 'choosing' to apply them? I think you have more to say from your own data than just adopting a term that doesn't adequately summarise the range of retreat behaviours you see. Regardless of the interpreted retreat rate terminology, I would suggest it's more interesting that in a trough system, commonly expected to host well-formed lineations and

retreat from wedge to wedge, here you have a record dominated by retreat landforms of various morphologies, marking different grounding line sedimentation processes and/or rates, different grounding line durations, and that overall the magnitude of the retreat events is rather small. The atypical landform assemblage for a trough setting is more interesting to explore than applying a conceptual model of rate that you can't say much about. *We see the reviewers point and appreciate the many good ideas. This part of the discussion chapter has been restructured and rewritten, now focusing more on our own results rather than terminology.*

Line edits

Title: ice-sheet is hyphenated here, but nowhere else in the text. *Corrected.*

Abstract

p1 line 10-12: re-consider the phrasing here in light of uncertainties/assumptions in calculations (see main comment in Discussion text). *Rephrased.*

p1 line 13-14: what evidence do you have that ice retreated directly across Store Koldewey Island and Germania Land? You can argue that at the ice sheet's maximum extent, ice was sufficiently thick to flow across this topography, but in a late stage of deglaciation? Isn't it more plausible that the high ground forced flow paths (and retreat) around the topography? The latter is in fact what you conclude. This part of the abstract should be revised. *We agree and have rephrased.*

Main text

p1 line 20: "…GIS is presently drained…" *Corrected.*

p2 line 14: explain why an absolute chronology is relevant to the previous sentence: chronology enables us to understand rates of change, while absolute ages let us tie retreat events in with external forcing (climate or ocean changes). *We have rephrased the sentence to make it clearer that in order to link retreat events with external forcing, an absolute chronology for the deglaciation is required.*

p2 line 17-19: do these cosmogenic dates record ice retreat, as in movement of the ice front landward of the island, or do they record ice sheet thinning? *We have rephrased and made it clear that the cosmogenic dates record ice front retreat at Store Koldewey Ø.*

p3 line 1: it deepens seaward but with bumps and dips along its length – it's not a smooth/steady deepening. *Rephrased.*

p3 line 5-8: while interesting, it doesn't seem relevant to the rest of the paper that this ice stream has displayed two recent surges. *We see your point and have removed this paragraph.*

p3 line 20: suggest ending this section with some sort of motivation statement for the rest of your work, or introductory statement to what you've done that builds on these previous studies that you mention. *Included.*

p3 line 33: suggest here you comment on which data have already been published in Laberg et al, and which are new (i.e. move from p5 line 36-38). There also is a bit of a methods gap between this paragraph and the next, where you explain GZW volume estimates, despite not having mentioned that you have / how you have recorded GZWs. I think a sentence or so stating that you've mapped landform outlines or crestlines is needed, and perhaps the basis for how you've interpreted the

environmental origins and how/why you've re-interpreted some earlier published ideas (e.g. assessment of size, shape, arrangement, sedimentary setting…). *These are many good suggestions that we appreciate! We have now included a paragraph in the Material and Methods chapter where we inform the reader on which data is new and which has previously been reported by Laberg et al., as well as how the classification of landforms was conducted.*

p3 line 35: what do you mean by 'box volumes'? You've assumed a rectangular cross-profile? On what basis? Is this more valid than an asymmetric triangle? *We have rephrased the paragraph, now explaining that the sediment volumes of the grounding zone wedges were calculated as trapezoid prisms, using a mean thickness and length obtained from the acoustic data.*

p3 line 38: you could refer to other works that have followed a similar approach… *References are now included.*

p4 line 2: you could note that this is a common problem in cold polar waters close to ice sheet grounding lines (relatively little life at the grounding line and low carbonate preservation subsequently (dissolution)) – plenty of other Greenland and Antarctica studies suffer the same limitations. *Good idea, this is now included.*

p5 line 6: 'more dominant component AT the expense of…' *Corrected.*

p5 line 24: refer to your sand-silt-clay data, specifically, to support that facies 1 is coarser than facies 2… although, looking at these plots, the coarser nature of facies 1 is not particularly convincing. *We see a small, but an overall coarsening in grain-size going from facies 2 to facies 1 due to the introduction of IRD.*

p5 line 26: suggest you note in the facies 1 and facies 2 observations that these two facies alternate in two of the cores – there may be more than one occurrence of each/either facies in the core succession. *Corrected.*

p5 line 35: both Fig 4 and Fig 5 should be referenced here, not only Fig 5. E.g. '…data from the middle (Fig 4) and outer (Fig 5) shelf of…' *Corrected.*

p6 line 14: what data did Laberg et al interpret wedges A-D from? Presumably not all multibeam, since you present a new block here? Seismic? *Laberg et al. interpreted the two outermost wedges (A and B) on available multibeam data, whilst the two innermost (C and D) were based on IBCAO 3.0.*

p6 line 17-18: can you give more information on how you calculated these volumes? Did you choose just one cross-profile per wedge, or several, and on what basis? From the multibeam, wedge C looks considerably larger than A, more comparable to B. *We have explained our calculations in the Methods chapter. A possible explanation for GZW C appearing larger than it is may be due to differences in both the scales and water depth legends between figure 4 and 5.*

p6 line 31-32: a figure and more detailed presentation and discussion is warranted here if there are genuinely two superimposed sets of moraines. I can't see that this is evident from either the multibeam panel or the mapped interpretation of moraines in Fig 5. On what basis do you interpret superimposition? If there is, in fact, one group that sits clearly on top of another, then we must interpret that there's been a readvance: you would have one retreat assemblage buried under another. And in that case, this finding would warrant further discussion. *We have specified in Fig. 5 where the superimposing recessional moraines occur, as well as including it in the Discussion chapter.*

p7 line 10: describe or explain why you find the sawtooth-like pattern incompatible with your earlier interpretation. *Rephrased.*

p7 line 14: '…identified along the … sidewalls' sounds like channels are running parallel to the walls of the trough. These simply occur at the periphery of your data coverage, and cut obliquely through other landforms. Suggest you rephrase. *Rephrased.*

p7 line 37: either break the sentence after the list of lithological units ('… and 1 (Fm(d)). It occurs at all…') or insert 'and' before 'it'. *Corrected.*

p8 line 10-13: suggest you switch these two sentences around, it flows more logically from the previous paragraph to begin with the till, and then what rests on top of the till. *Good idea!*

p8 line 25: 1000-1500m thick, or high? (i.e. surface altitude or ice thickness?) *Rephrased to ice thickness.*

p8 line 30: since Storstrømmen is an outlet of the contemporary NEGIS, it sounds strange to talk about this as a 'similar flow feature'. Can you instead emphasise here that the disregard for topography appears (from your results) to be a characteristic of both the palaeo and contemporary NEGIS? And be specific about exactly where this independence from topographic steering occurs within the NEGIS today. *Publications indicate that the contemporary NEGIS appears to have similar characteristics as pure ice streams (Fahnestock et al., 1993; Sachau et al., 2018).*

p8 line 31: 'a palaeo-ice stream' – can you be specific, which one? *We have clarified that it is the Maskwa paleo-ice stream within the Laurentide Ice Sheet.*

p8 line 34-38: do Arndt et al propose a more restricted ice margin as well as a limited drainage basin (supply), or do they also suggest shelf-edge glaciation? If they envisage shelf-edge glaciation, then your counter-argument ('we have evidence of shelf-edge glaciation') isn't really sufficient. *Arndt et al. do not suggest any former position of the ice margin explicitly; however, they suggest that Store Koldewey Trough probably was not eroded by an ice stream during multiple glaciations. They further propose that major parts of the ice on Germania Land drained northwards into Norske Trough and to Dove Bugt in the south. If so, we believe that there would not have been sufficient amount of ice left to fill Store Koldewey Trough all the way to the shelf edge during LGM.*

p8 line 37: can you be more specific about the volume of ice required, and compare to what Germania Land could sensibly supply? *We have provided information on possible ice volume on Germania Land, but not for the trough during LGM. However, we have included numbers on the area and depths of the trough.*

p8 line 42-45: I don't think these opening sentences really add anything useful. *Removed.*

p9 line 5: the phrasing here makes it sound like Store Koldewey also has a reverse slope. The fact that it doesn't is surely an important reason for any contrast in behaviour. You set up a 'problem' here that isn't really one. I would revise this paragraph as a commentary on the rather uncommon situation of having a seaward dipping trough that has led to a rather stable retreat pattern, supplemented by local trough shallow/narrow points, rather than making this more of a puzzle than it really is. *We see the reviewers point and have rewritten the paragraph.*

p9 line 13-14: what do you mean by 'had a more dynamic response to…'? Rephrase to say something direct, this is vague. *Removed.*

p9 line 16: 'local trough geometry' (typo, through) *Corrected.*

p9 line 22: where are the other 2 wedges that Batchelor & Dowdeswell find here, and why do you not include those in your reconstructions? *We find it unclear where Batchelor and Dowdeswell (2015) have identified the two remaining GZWs and have therefore not included them in our reconstructions.*

p9 line 20-27: this paragraph doesn't seem to go anywhere. What do you interpret to be the significance in the number of wedges recorded in different troughs? Are the single wedges in Norske & Westwind Troughs at the shelf break? What would be the implications for your work if these formed during the Younger Dryas? *We have removed this paragraph.*

p9 line 35: which West Antarctic ice stream? Or do you mean ice sheet? *Corrected to Whillans Ice Stream, West Antarctica.*

p9 line 37: these values use the upper sediment flux rate. Using the lower flux would give you an order of magnitude longer formation time, i.e. 1300, 7400 and 1500 years. Is there enough time available for retreat across this shelf, with those standstill durations? See also main comment about sediment flux assumptions. *We have now included the lower flux rate in our discussions.*

p9 line 40: this passage must better reflect the debate about whether recessional moraines can be interpreted as annual or not. *Included.*

p9 line 50-52: this sentence either could be removed (since you don't put it in the context of your results) or should reflect the vast literature on grounded to open marine (deglacial) sedimentological facies. Picking a random three papers that have studied this succession is rather meaningless. *Removed.*

p9 line 6: masks *Corrected.*

p9 line 7-9: Prothro et al (Marine Geology) discuss the distance for rainout of basal debris distal to the grounding line – they find it to be extremely short. *Ok.*

p9 line 25: it is intriguing that the sediment drape only occurs across the inner-middle shelf, and not the outer parts that deglaciated first. Why do you think this is? Is this a supply or a preservation/deposition question? *We have provided a possible explanation for this (current winnowing, leading to non-preservation of fine sediments).*

p9 line 30: what is the significance of facies 1 & 2 alternating in the two outer cores? How does this affect your environmental interpretations here? *Included.*

p9 line 37: would reduced sea ice not allow more icebergs to access the area? Or, conversely, expanded sea ice would limit access of icebergs and deposition of IRD? *Reconstructions of sea-ice concentrations and IRD data on the East Greenland shelf reveal that a reduced sea-ice cover during the Early Holocene was accompanied by low IRD concentrations (e.g. Müller et al. 2012).*

Figure 1: label the profile shown in panel B on the dashed white line in panel A (instead of or as well as writing in the caption). *Ok.*

Figure 5: could you make the colours for sawtooth moraines and iceberg ploughmarks more distinct from one another? *Ok.*

**Changes made to the manuscript:**

In the following revised version of the manuscript, we have added and re-written paragraphs, as well as corrected misspellings and rephrased sentences.

Methods:

All radiocarbon dates cited in the manuscript were calibrated using the CALIB 8.20 software, applying the Marine20 calibration curve.

We have provided information on how the mapping and classification of landforms were conducted, as well as our calculations on grounding-zone wedge sediment volumes.

Results:

In chapter *4.1 Lithostratigraphy of the uppermost trough strata*, we have added some alternative interpretations, in addition to adding additional references to our interpretations, as requested by the reviewer.

Our interpretations of mega-scale glacial lineations in the trough has been slightly modified – we have now opened for the streamlined landforms in the middle trough to be interpreted as "glacial lineations".

Discussion:

We have restructured and rewritten parts of the Discussion chapter, focusing more on our own results, as well as acknowledging the uncertainties within our interpretations and estimates of retreat rate/duration of standstills.

Finally, we added a paragraph with "Acknowledgements".

[revised manuscript text omitted]

5 The complex glacial landform assemblage in Store Koldewey Trough, comprising transverse wedge- and ridge systems, reflects to a large degree the dynamic retreat of the ice margin during the deglaciation. The types of landforms and their spatial distributions can be attributed to the overall seafloor topography of the trough, with a seaward dipping bed slope, supplemented by local pinning points related to trough shallowing and/or narrowing. The resulting deglacial dynamics was characterized by several periods of stabilization and readvances of the

10 grounding line in Store Koldewey Trough during overall retreat. In contrast,  many paleo-ice streams on other glaciated continental shelves with  landward dipping beds  have experienced a lift-off from the seafloor and an initial rapid retreat due to sea-level rise, e.g. Norske Trough (Arndt et al., 2017), Amundsen Sea in West Antarctica (Smith et al., 2011) and NW Fennoscandian Ice Sheet (Rydningen et al., 2013), the ice stream in Store Koldewey Trough repeatedly stabilized and readvanced, interrupting the deglaciation.

~~. A combination of several factors may have preconditioned this, and led to the complex geomorphology in Store Koldewey Trough: i) local highs in the overall shallowing landward seafloor profile may have provided pinning points, causing ice stabilization and promoting longer stillstands during the deglaciation; ii) trough narrowing towards the coast may have increased lateral stress on the retreating ice margin, thus slowing down/stabilizing ice~~

20 ~~flow; iii)) the GIS in Store Koldewey Trough possibly had a more dynamic response to the changing climatic and oceanographic conditions compared to troughs of similar dimensions elsewhere on the NE Greenland Margin. The grounding zone wedges in Store Koldewey Trough occur in areas of relatively marked decreased water depths and reductions in trough width (Fig. 1), indicating that local through geometry led to a repeated re-stabilization of the grounding line.~~

[revised manuscript text omitted]

---

## Author Response (AR3)

Dear prof. Stokes!

Thank you for accepting our manuscript "Last Glacial ice sheet dynamics offshore NE Greenland – a case study from Store Koldewey Trough". We have provided the technical corrections as requested, as well as corrected some misspellings. All changes are marked below.

Kind regards on the behalf of the authors,

Ingrid Leirvik Olsen

[revised manuscript text omitted]